# Mechanism of activating mutations and allosteric drug inhibition of the phosphatase SHP2

Ricardo A.P. Pádua[1], Yizhi Sun [1,2], Ingrid Marko[1], Warintra Pitsawong[1], John B. Stiller[1], Renee Otten [1] & Dorothee Kern[1]

Protein tyrosine phosphatase SHP2 functions as a key regulator of cell cycle control, and activating mutations cause several cancers. Here, we dissect the energy landscape of wild-type SHP2 and the oncogenic mutation E76K. NMR spectroscopy and X-ray crystallography reveal that wild-type SHP2 exchanges between closed, inactive and open, active conformations. E76K mutation shifts this equilibrium toward the open state. The previously unknown open conformation is characterized, including the active-site WPD loop in the inward and outward conformations. Binding of the allosteric inhibitor SHP099 to E76K mutant, despite much weaker, results in an identical structure as the wild-type complex. A conformational selection to the closed state reduces drug affinity which, combined with E76K's much higher activity, demands significantly greater SHP099 concentrations to restore wild-type  activity levels. The differences in structural ensembles and drug-binding kinetics of cancer-associated SHP2 forms may stimulate innovative ideas for developing more potent inhibitors for activated SHP2 mutants.

[1] Howard Hughes Medical Institute, Department of Biochemistry, Brandeis University, Waltham, MA 02454, USA. [2] Present address: Department of Cancer Biology and Cell Biology, Dana-Farber Cancer Institute and Harvard Medical School, Boston, MA 02215, USA. These authors contributed equally: Ricardo. A. P. Pádua, Yizhi Sun.  Correspondence and requests for materials should be addressed to D.K. (email: dkern@brandeis.edu)

The development and propagation of proliferative diseases can most often be ascribed to genetic errors that disturb the finely tuned cell signaling pathways. Treatment remains difficult due to the multiplicity of shared protein folds, leading to toxic off-target effects during orthosteric chemotherapy. Instead, more selective and effective drugs can be produced by targeting the allosteric network of proteins, which, through subtle, epistatic evolution, have developed uniquely, unlike conserved active sites. Recently, an allosteric inhibitor (SHP099) was developed for the nonreceptor protein tyrosine phosphatase SHP2[1,2], a fundamental enzyme for cell cycle control, and the root of many pathologies such as LEOPARD syndrome, Noonan syndrome (NS)[3–5], and juvenile myelomonocytic leukemia[6,7].

The full-length, wild-type SHP2 (FL-WT) contains three domains: a protein tyrosine phosphatase domain (PTP) and two preceding Src homology 2 domains (N-SH2 and C-SH2)[8,9]. Unperturbed, SHP2 exists in an auto-inhibited state with the N-SH2 domain docked into the catalytic cleft of the PTP[8]. The binding of a phosphotyrosine peptide to the opposing face of the N-SH2 domain exposes the catalytic cleft to substrate and activates the system[10]. In diseases caused by SHP2, mutations are often observed at the N-SH2/PTP interface (e.g., E76D/E76K), resulting in constitutively active protein and abnormal cellular proliferation[11–13]. The recently developed inhibitor, SHP099, allosterically closes the protein and deactivates SHP2 by stabilizing the N-SH2/PTP interaction. Although SHP099 exhibits nanomolar affinity for wild-type SHP2 and is a possible cure for diseases caused by SHP2 upregulation[14], it remains unclear whether or not it can be a potent treatment against diseases caused by activating mutations in SHP2.

In this study, we utilized a combination of nuclear magnetic resonance (NMR) spectroscopy, X-ray crystallography, small-angle X-ray scattering (SAXS), enzyme kinetics, isothermal titration calorimetry (ITC), and stopped-flow kinetics to show that SHP2 exists in a dynamic equilibrium between a closed state (inactive) and an open state (active). The oncogenic mutations of SHP2 (FL-E76D and FL-E76K) were characterized and found to shift the open/closed equilibrium toward the open species. Additionally, we describe two structural features for SHP2: (i) the structure of the open, active conformation of SHP2 with a PTP/C-SH2 interface that is vastly different from the interface of the inactive state, and the N-SH2 detached from PTP; and (ii) direct detection of the inward conformation of the active-site WPD (for Trp-Pro-Asp) loop (WPD-in) in the ligand-free protein that was previously seen only in an outward conformation (WPD-out) in SHP2. We further show that the SHP099 inhibitor binds via a pure conformational selection mechanism, associating only with the closed state, and, therefore, the oncogenic mutations vastly reduce the inhibitor affinity.

## Results

### Differences in structural ensembles between WT and E76K-SHP2.
Although the auto-inhibited SHP2 structure is well established[8,15–17], the contrasting active form, commonly referred to as the open conformation, has remained obscure. As wild-type SHP2 is presumed to sample the open conformation infrequently, we investigated an NS/leukemia-associated mutant, E76K. As one of the most active disease mutants of SHP2, the E76K mutation disrupts the N-SH2/PTP domain interface, resulting in accelerated enzymatic turnover and, presumably, a more populated open state[12,18]. To gain structural knowledge of how the open state differs from the auto-inhibited state, we began by acquiring [$^1$H-$^{15}$N]-TROSY-HSQC NMR spectra of full-length, wild-type SHP2 (FL-WT; residues

1–529, lacking the C-terminal tail) and the E76K mutant (FL-E76K). With the hypothesis that SHP2 is regulated by a classical allosteric equilibrium between a closed, inactive (I) and open, active (A) conformation[8,19], chemical shift differences between wild-type and E76K SHP2 can provide atomistic information on structural differences in addition to thermodynamics of the equilibrium. A comparison of both spectra shows significant differences in chemical shifts for most of the dispersed cross peaks (Fig. 1a), indicative of a global conformational change between wild-type and mutant form. To spatially characterize the differences in chemical shift, backbone assignments for wild-type (FL-WT) and the PTP domain were obtained from triple-resonance experiments, and for mutant (FL-E76K) by a modular-domain comparison, minimal distance approach (Supplementary Fig. 1). Plotting the chemical shift differences between FL-WT and FL-E76K on the closed structure illustrates that most prominent changes occur along the N-SH2/PTP and C-SH2/PTP interfaces (Fig. 1b), consistent with the notion that FL-E76K samples a more elongated form with the N-SH2 domain detached from the PTP.

To validate such detached model for the open state, full-length spectra were compared with those obtained for tandem-SH2 domain constructs (WT/E76K; residues 1–217) (Fig. 1c, d). For residues within the N-SH2 domain, chemical shifts in the tandem-SH2 construct overlay well with FL-E76K, indicative of a "free" domain in the full-length mutant form. In contrast, the corresponding C-SH2 chemical shifts are largely different between tandem-SH2 and full-length proteins, with greater similarity to a "free" C-SH2 domain appearing in the FL-WT. Rather than a completely detached domain model, the open state appears to possess a dissociated N-SH2 domain with a different C-SH2/PTP interface (Fig. 1e).

### Structural characterization of the open conformation.
To further understand the molecular basis of this different C-SH2/PTP interaction, we attempted to capture the open conformation by crystalizing FL-E76K, but failed. We reasoned that the flexibility of the released N-SH2 domain caused structural heterogeneity precluding crystallization. Therefore, we used a truncated SHP2 lacking the N-SH2 domain (ΔN-SH2; residues 104–529)[20]. As a single point mutation at the N-SH2/PTP interface (E76K) results in opening, we predicted that the ΔN-SH2 would resemble the completely open structure, albeit crystallizable. In agreement with our expectation, cross peaks in the [$^1$H-$^{15}$N]-TROSY-HSQC spectrum of ΔN-SH2 are nearly superimposable with the corresponding SHP2-E76K peaks (Fig. 2a, Supplementary Fig. 3), assuring that the ΔN-SH2 construct mimics the full-length, open conformation (Fig. 2b). We then crystallized and solved the ΔN-SH2 structure at 2.9 Å. It contains four molecules in the asymmetric unit with subtle rotational differences, indicative of an ensemble of open states (PDB 6cmq, Fig. 2c). Remarkably, each structure shows an approximate 120° rotation of the C-SH2 and PTP domain with respect to the auto-inhibited structure (Fig. 2d). The hinge motion of the C-SH2 domain is achieved by changes in the main-chain torsion angles of residues L216, N217, and T218 in the linker region. A rotation, and not a detachment, of the C-SH2 domain results in a distinct C-SH2/PTP interface, which is stabilized by several electrostatic interactions within the C-SH2/PTP interdomain linker (Fig. 2d). These contacts reconcile the large chemical shift differences observed in the C-SH2 domain for FL-E76K compared with FL-WT and tandem-SH2 construct (Fig. 1). Additionally, a comparison between the ΔN-SH2 structure with the open conformation of SHP1[21] reveals that the C-SH2 interface is shared between these proteins (Fig. 2e). In both structures, the C-SH2 rotation moves the N-SH2 away from the

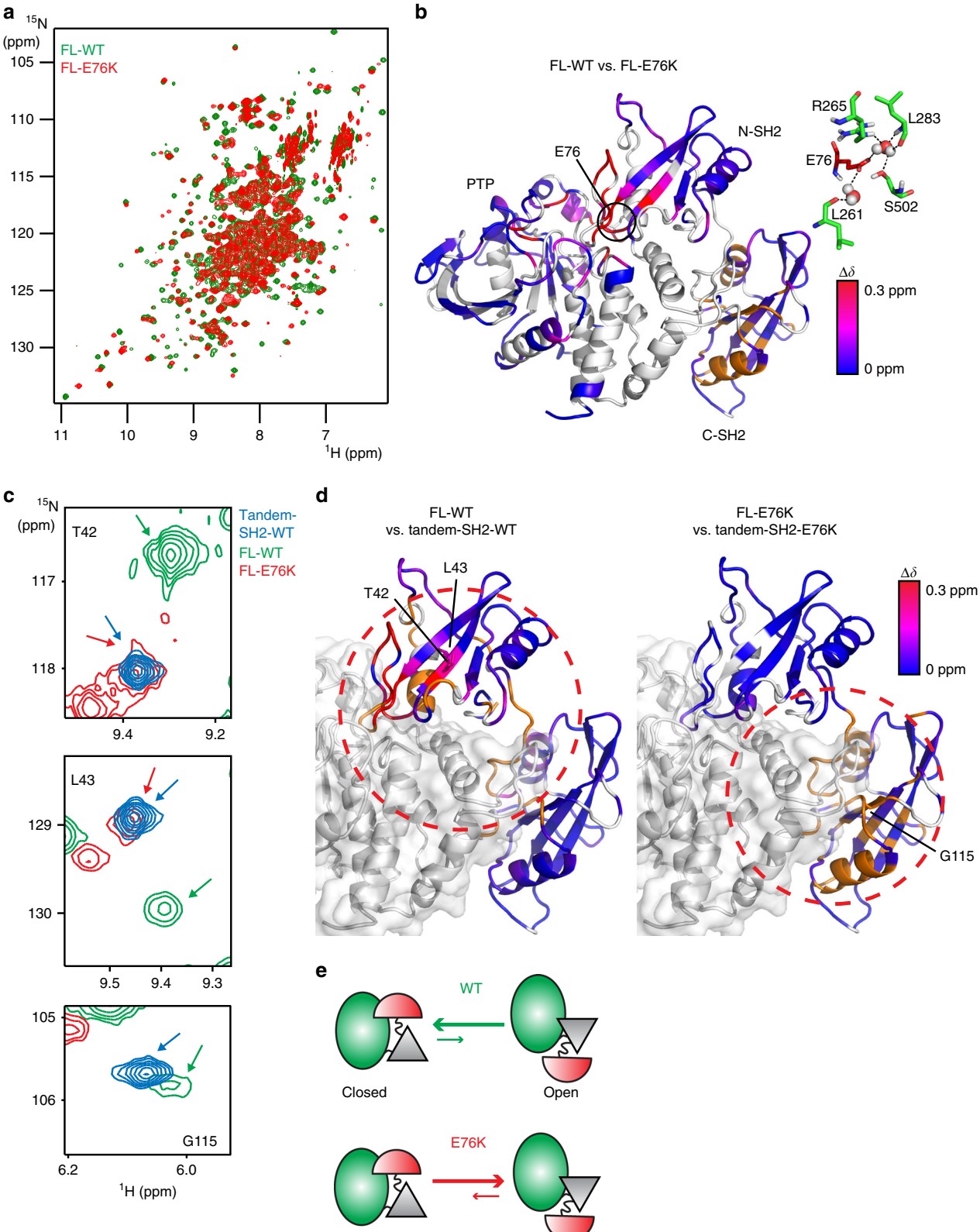

PTP domain's active site. In the open SHP1 structure, the N-SH2 docks along the backside of the PTP, potentially stabilizing the open conformation (Fig. 2e). Although these contacts in the SHP1 crystal structure appear resolute[21], analysis of the NMR chemical shifts in SHP2 for residues in the N-SH2 of FL-E76K shows no notable differences when compared with the tandem-SH2 construct (Fig. 1d). The lack of chemical shift perturbations indicates that in solution the N-SH2 lacks contacts with the PTP in SHP2.

To verify this "free" N-SH2 hypothesis, SAXS experiments were performed on the FL-E76K and FL-WT proteins. In confirmation of our NMR data, the calculated structure in which

**Fig. 1** NMR chemical shift perturbations caused by the activating disease mutation E76K of SHP2. **a** Superposition of the [$^1$H-$^{15}$N]-TROSY-HSQC spectra of FL-WT and FL-E76K. **b** Chemical shift differences of **a** were plotted on the closed, inactive crystal structure of FL-WT (PDB 4dgp[15]). Unassigned, overlapping, or prolines residues are shown in gray; the orange color represents residues that experience chemical shift perturbations, but the precise value of Δδ is unknown because assignments are only available for one of the proteins. The interactions between E76 and residues in the PTP domain are shown on the right, with water molecules displayed as spheres. **c** Selected regions of the overlaid spectra of FL-WT, FL-E76K, and tandem-SH2-WT (see also Supplementary Fig. 2). **d** A red, dotted circle highlights major chemical shift perturbations in both SH2 domains for the isolated tandem-SH2 versus FL-WT (left), and only in the C-SH2 domain for the same comparison of E76K (right). Color coding is the same as in panel **b**. **e** Schematic representation of the open/closed conformational equilibrium of FL-WT and FL-E76K (N-SH2, red; C-SH2, gray; PTP, green)

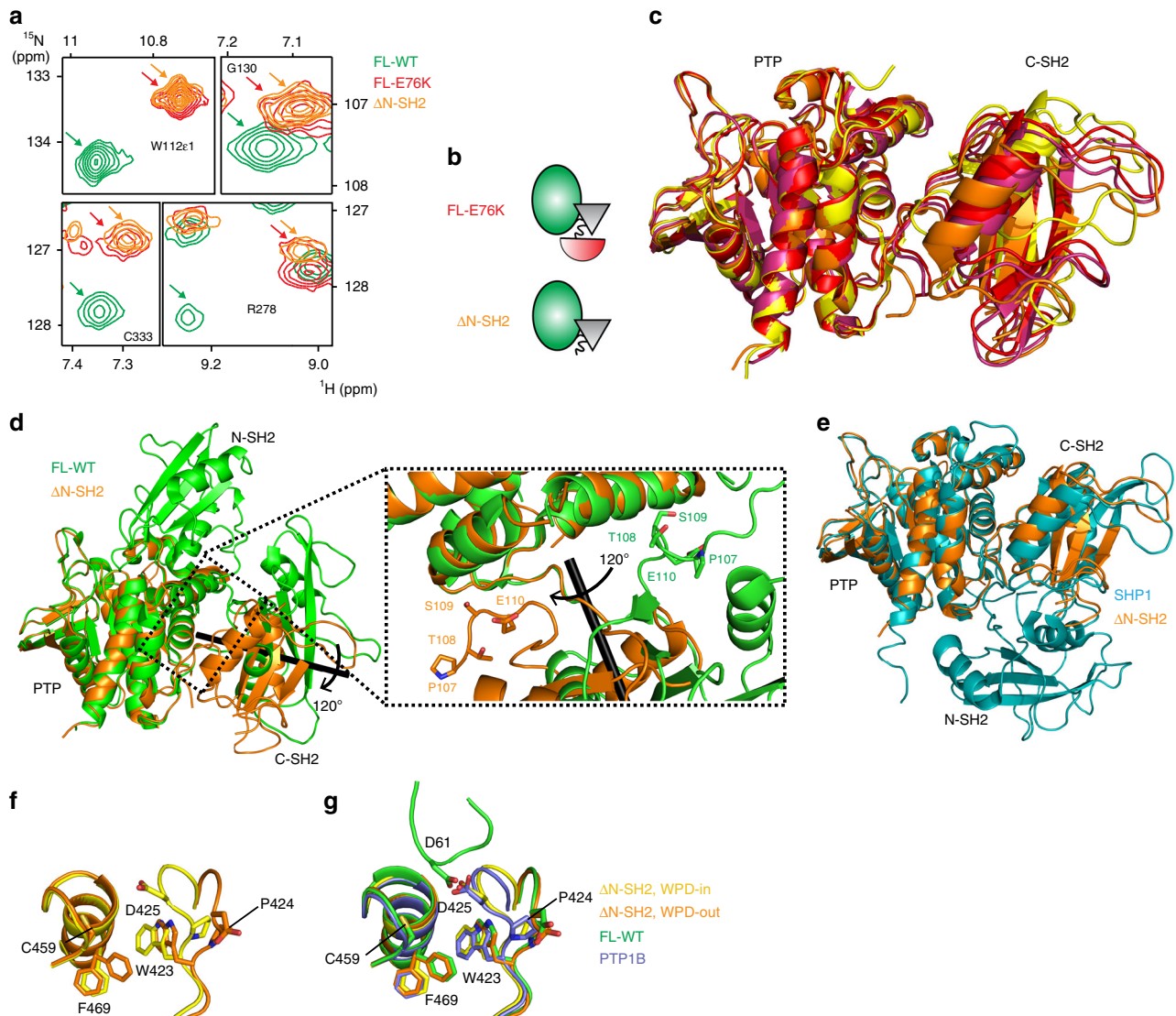

**Fig. 2** Structure of the open active state of SHP2. **a** Chemical shifts of ΔN-SH2 are nearly identical to FL-E76K indicating that removal of the N-SH2 mimics the structure of FL-E76K, shown as cartoon in **b**. **c** The four molecules found in the asymmetric unit of the ΔN-SH2 crystal structure (purple, red, orange, yellow) were superimposed using the PTP domain as reference to reveal an ensemble of orientations adopted by the C-SH2 domain. **d** When compared with FL-WT (green), the C-SH2 domain of the ΔN-SH2 structure (orange) is rotated by 120° (axis in black). This movement positions the SH2-SH2 linker in between the PTP domain and the C-SH2 as exemplified by residues P107-E110 (inset). **e** The C-SH2 domain in the ΔN-SH2 open structure (orange) adopts a similar conformation to the C-SH2 in the open state of SHP1 (PDB 3ps5, teal)[21]. **f** Zoom-in of the active site of ΔN-SH2: conformational sampling of the WPD-in (yellow) and WPD-out states (orange, seen in all previous SHP2 structures). **g** The N-SH2 loop from FL-WT (green) can only dock into the PTP domain when the WPD is in the out conformation. The WPD-in conformation of ΔN-SH2 crystal structure is identical to the one reported for the PTP1B (PDB 1sug, blue)[24]

the N-SH2 is detached from the PTP (Supplementary Fig. 4a, b) fits better to FL-E76K SAXS data ($\chi = 1.59$) compared with the crystallized "open conformation" of SHP1 ($\chi = 2.82$) (Supplementary Fig. 4c, d). FL-E76K in solution has a bigger radius of gyration ($R_g = 29.10 \pm 0.62$ Å) and maximum dimension ($D_{max}$

$= 95.3$ Å) than FL-WT ($R_g = 26.85 \pm 0.43$ Å, $D_{max} = 88.7$ Å, Supplementary Fig. 4e-g), which is consistent with previously reported changes in SHP2 SAXS structural parameters caused by other gain-of-function mutations[17]. In addition, the scattering experiments on FL-WT agree reasonably well with closed crystal

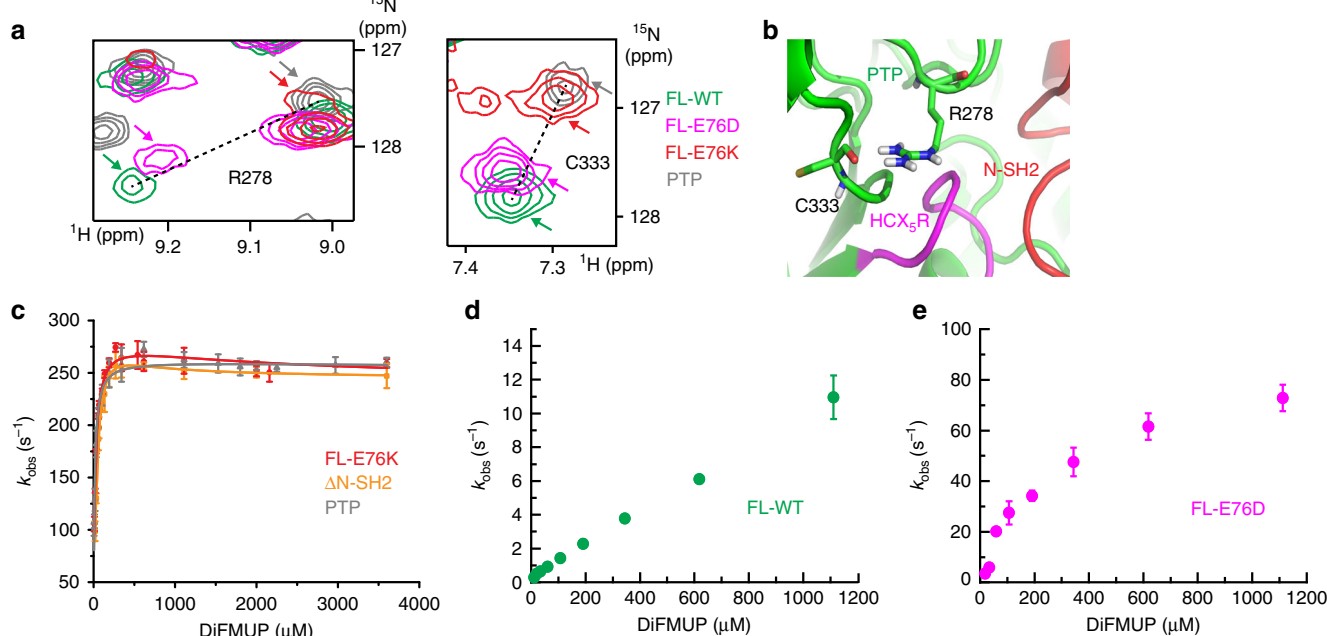

**Fig. 3** Open/closed equilibrium of WT and SHP2 mutants correlates with enzymatic activity. **a** Selected region of the [$^1$H-$^{15}$N]-TROSY-HSQC spectra (residues R278 and C333 in the PTP domain) shows linear shifting of cross peaks from PTP via FL-E76K and FL-E76D to FL-WT. **b** R278 and C333 are shown in stick representation, showing their spatial proximity to the active site of the catalytic domain (the HCX$_5$R motif) and the N-SH2 domain. **c–e** Phosphatase activity as a function of substrate (DiFMUP) concentration measured at 35 °C. **c** The profiles for PTP, ΔN-SH2, and FL-E76K are within experimental error and were fit to the Michaelis–Menten model with partial substrate inhibition (Eq. 1, fit parameters in Supplementary Fig. 7b). **d**, **e** FL-WT and FL-E76D show much less activity combined with a weaker apparent $K_M$. Uncertainties represent the standard deviation of the sample ($n = 3$)

structure ($\chi = 1.92$, Supplementary Fig. 4e, f), but not with the open model for FL-E76K SHP2 ($\chi = 7.34$) or SHP1 ($\chi = 4.13$). Taken together, the NMR, crystallographic, and SAXS data indicate that the open form of SHP2 involves dissociation of the N-SH2 from the PTP domain through rotation of the C-SH2 domain acquiring a more elongated shape. The translocation of the N-SH2 from the active site allows for substrate binding and, therefore, opens the door for catalysis.

Although forming an accessible active site is a prerequisite for catalysis, it does not represent the only dynamic process in SHP2's catalytic mechanism. For example, the WPD loop in PTP1B was found to undergo exchange moving in and out of the catalytic site[22,23]. Upon substrate binding, the WPD-in state places residue D425 close to the phosphoryl-aryl group. During turnover, D425 plays two critical roles: (1) protonating the substrate phosphoryl group for nucleophilic attack by the active site cysteine and (2) deprotonating a water molecule for phosphocysteine hydrolysis. After the product is formed, the WPD loop moves away from the catalytic site (WPD-out conformation)[24]. In all SHP2 structures to date, the WPD loop is found in the out state, distant from the catalytic site[8,15]. However, in our four ΔN-SH2 molecules we observe both states (Fig. 2f). For chains A and D, the WPD loop is in the in state, which is comparable to that found in the homologous PTP1B structure (PDB 1sug)[24]. The WPD loop (chains B and C) is displaced 10 Å from the surface of the protein (Fig. 2f) to the out conformation, and overlays perfectly with FL-WT structure (Fig. 2g). In the auto-inhibited structure, the interaction of N-SH2 with the PTP domain sterically prevents substrate binding and drives the WPD to the out state (Fig. 2g). Conformational flexibility observed here for ΔN-SH2 parallels recent findings for PTP1B, which, through the use of room temperature crystallography, was found to sample both states of the WPD loop in a temperature-dependent manner[25]. Finally, we observe correlated

movement between the displacement of W423 of the WPD loop and the rotameric state of F469, located below the catalytic cysteine (C459). Within the WPD-in state, this aromatic reorganization further primes the active site for substrate binding and turnover.

**The active/inactive equilibrium that controls SHP2 activity.** Our results indicate that destabilization of the closed conformation and, consequently, an increased open population, is the principal mechanism of achieving faster turnover in the cancer-associated mutants of SHP2. To test this hypothesis further, we expressed and purified a second oncogenic mutant, E76D. Unlike the charge inversion of E76K, the E76D mutant form maintains wild-type's negative charge, but shortening of the side chain disrupts a hydrogen bonding network between residue 76 in the N-SH2 and the PTP residues R265 and S502 (see below). Acquisition of a [$^1$H-$^{15}$N]-TROSY-HSQC spectrum of FL-E76D reveals a hybrid profile with resonances in the PTP appearing in-between cross peaks for FL-WT and FL-E76K (Fig. 3a, b). Linearity between many resonances in the three full-length proteins and the individual PTP construct suggests that SHP2 exchanges rapidly between the inactive (closed) and active (open) conformations. Additionally, the observed linearity implies that the chemical shifts of residues in the N-SH2 and PTP domains report directly on the populations of the open and closed states for both WT and mutant proteins. The PTP domain and its chemical shifts are a true representation of the completely open form; however, a respective closed state is lacking. Nevertheless, chemical shift comparison assuming that FL-WT reports primarily on the closed species would suggest the population of the open state is >90% for FL-E76K and >10% for FL-E76D.

With these populations in mind, enzymatic phosphatase activity was measured using the synthetic DiFMUP substrate

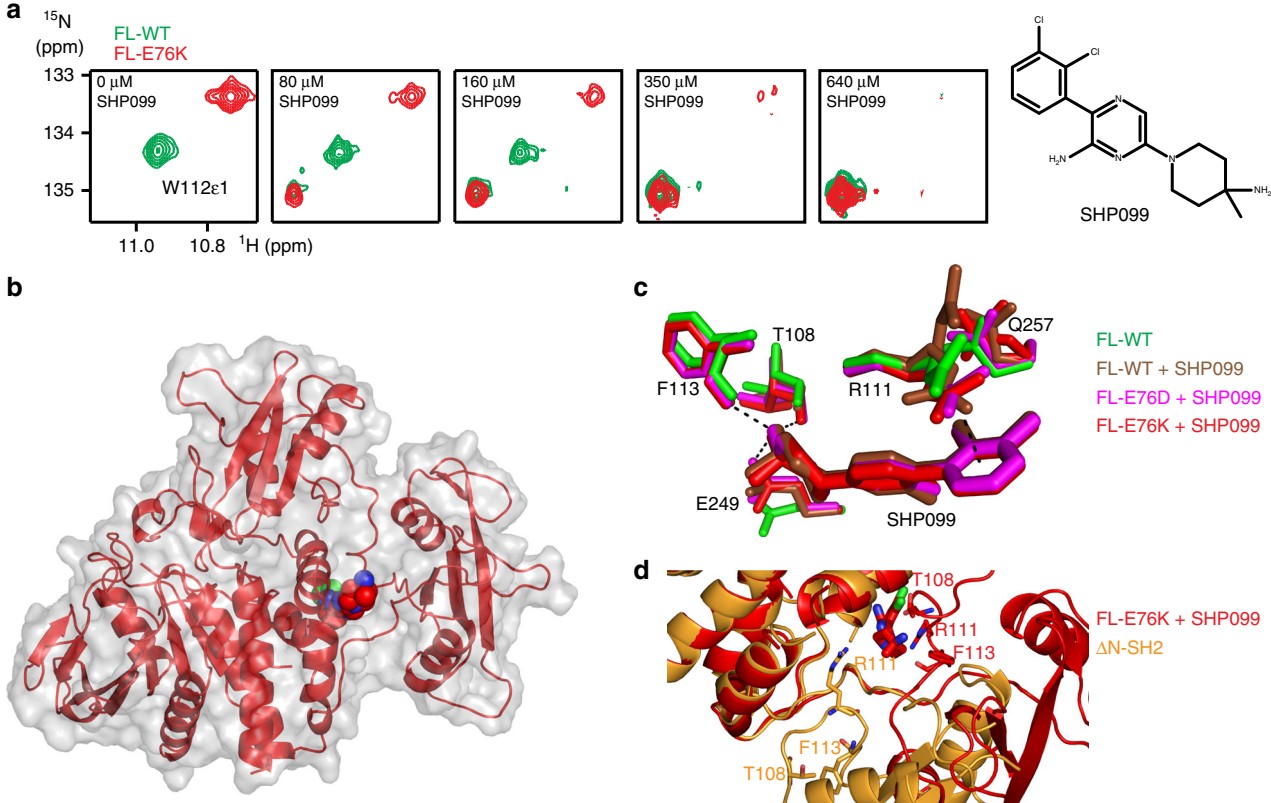

**Fig. 4** NMR and crystallographic analyses of the binding of SHP099 to SHP2 variants. **a** Representative chemical shift changes of W112ε1 as seen for many residues away from the site of mutation upon binding of increasing amounts of SHP099 to 340 and 320 μM FL-WT and FL-E76K, respectively. The chemical structure of SHP099 is shown on the right. **b** Surface representation of FL-E76K bound to SHP099 (in Van der Waals representation; PDB 6cms) reveals a closed SHP2 conformation very similar to FL-WT bound to SHP099[1]. **c** Superposition of the drug in the binding pocket of FL-WT and different mutant forms. The positions of residues important for SHP099 binding are already sampled in ligand-free structures of FL-WT (PDB 5i6v[17]). **d** Superposition of the ΔN-SH2 structure (orange, PDB 6cmp) with FL-E76K bound to SHP099 (red, PDB 6cms), using the PTP domain as reference. The rotation of the C-SH2 in the open conformation (orange) displaces residues T108, R111, and F113 from the SHP099 binding pocket

for the PTP domain, FL-E76K, ΔN-SH2, FL-E76D, and FL-WT SHP2 proteins. To avoid inner-filter effects at high DiFMUP concentrations, product formation was measured via absorbance rather than fluorescence as reported prior[2,17]. In agreement with the populations extracted from chemical shifts, FL-E76K has a nearly identical activity profile to the PTP and ΔN-SH2 constructs (Fig. 3c). We note that the activity did not follow a standard Michaelis–Menten model (see Supplementary Fig. 7a), but exhibited partial substrate inhibition that was modeled to a substrate sequential binding scheme (Supplementary Figs. 7b and 10a). In contrast to the turnover rates of FL-E76K, full-length wild-type and the E76D mutant are significantly inhibited with about a 23- and 3.6-fold reduction of $k_{obs}$ at 1.1 mM substrate, respectively (Fig. 3d, e). FL-WT and FL-E76D not only have a reduced $k_{obs}$ at all substrate concentrations, but also a weaker observed $K_M$ of the substrate (cf. Fig. 3c–e).

**Structure of SHP2-E76K bound to allosteric inhibitor SHP099.** Recently, a nanomolar, allosteric inhibitor, SHP099, was synthesized and found to stabilize the auto-inhibited, closed form of WT SHP2[1]. The co-crystal structure revealed that closure was induced through binding at a central cavity, formed at the interface of the N-SH2, C-SH2, and PTP domains. As oncogenic mutations characterized here (E76K and E76D) appear to destabilize the closed conformation, it remains an open question how SHP099 interacts with the modified energy landscapes. As a primer, NMR titration experiments of SHP099 to FL-WT protein were performed, and binding occurs in the slow NMR time regime

indicative of a slow off rate (Fig. 4a). Consistent with our findings that FL-WT is primarily closed and FL-E76K primarily open, chemical shift changes upon SHP099 binding are larger for the mutant relative to WT (Supplementary Fig. 5a, b). Comparison of inhibitor-bound spectra of FL-E76K and FL-WT found remarkable similarity for many residues, notable differences are only observed along the N-SH2/PTP interface (Supplementary Fig. 5c, d). The convergence of chemical shifts between WT and mutant proteins indicates that both forms close upon inhibitor binding. To confirm this hypothesis, we crystallized both SHP2 oncogenic driver mutants, E76K and E76D, with SHP099 and indeed found the closed structure of SHP2/inhibitor complex nearly identical to the FL-WT[1] (Fig. 4b, c).

**Mechanism of allosteric inhibition of SHP2 forms by SHP099.** From the inhibitor-bound crystal structures, one could conclude that the drug induces closure after binding, with the SHP099 bound open/closed equilibrium skewed heavily toward the auto-inhibited state. The first hint that such a mechanism is incorrect comes from a structural inspection of the drug-binding pocket. Most conformations of the residues involved in SHP099 binding can already be seen in ligand-free FL-WT crystal structures, arguing against an induced-fit mechanism[17] (Fig. 4c). In addition, the N-SH2 is far removed from the binding pocket in the open structure and reorientation of the C-SH2 completely disrupts the binding pocket (Fig. 4d). Also, we see no NMR chemical shift changes for ΔN-SH2 when titrating in SHP099. An alternative binding mechanism can be conformational selection, in which the

inhibitor binds solely to the closed population of SHP2. The distinction of whether the conformational change happens before (i.e., conformational selection) or after (i.e., induced fit) drug binding can only be made by measuring the flux[26–29].

In agreement with a conformational selection model, we measure a much weaker affinity of SHP099 to FL-E76K than to FL-WT, which parallels the population difference of the closed form (Fig. 5a, b).

To further substantiate the conformational selection binding mechanism, we investigated SHP099 binding to two additional SHP2 mutants that were originally designed as catalytically dead mutants, with the active site C459 changed to either Ser[30] or Glu[31]. Against our expectations, FL-C459S was found to be predominantly in an open conformation, with chemical shifts comparable to FL-E76K (Fig. 5c and Supplementary Fig. 6a). The equivalent mutation in PTP1B, C215S, has been reported to cause drastic changes in the orientation of the phosphate-binding loop (P-loop) likely due to the change of a negative charge from the thiolate anion to a neutral hydroxyl group[32]. If a negative charge is indeed crucial for the closed form, the C459E mutant would more properly mimic FL-WT. In agreement with our expectation, FL-C459E is indeed primarily closed as shown by NMR (Fig. 5c and Supplementary Fig. 6b) and crystallography (Supplementary Fig. 6c, d) making C459E a superior "dead mutant" that better resembles the closed conformation of wild-type SHP2. Strikingly, SHP099 affinities to all FL-SHP2 forms correlate well to the corresponding equilibria between open and closed states: FL-WT, -76D, and -C495E are predominantly in the closed conformation with tight affinities for SHP099; FL-E76K and -C459S with predominantly open conformations have much weaker affinities (Fig. 5d, Supplementary Fig. 6e–g).

Furthermore, as the binding of SHP099 shifts NMR cross peaks of FL-WT and FL-E76K to similar positions for many resonances (Fig. 5e, Supplementary Fig. 5), the SHP099-bound spectra can be considered to originate from a pure closed sample and the open/closed equilibrium can be quantified through chemical shift comparison (Fig. 5e). Using the drug-bound states as the chemical shift for closed, and the PTP domain for open, we find that FL-WT, -E76D, and -E76K sample the open, active conformation at $10.4 \pm 1.1\%$, $23.0 \pm 3.0\%$, and $95.8 \pm 1.7\%$, respectively. Considering these populations along with the closed SHP2/SHP099 complexes (Fig. 4b, c) imply that destabilization of the closed conformation is the primary mechanism for activating full-length SHP2 (Fig. 5f).

**Slow opening/closing rates and regulation of SHP2 activity.** Finally, differences in SHP099 binding between FL-WT and FL-E76K were studied kinetically using stopped-flow fluorescence to unambiguously validate a conformational selection mechanism[26–29]. The importance of drug-binding kinetics, particularly the drug on-target life-times, has been well recognized[26,33–35]. SHP099 binding to FL-WT displays double-exponential kinetics (Fig. 6a) that can be assigned to the fast association step with a bimolecular rate constant, $k_{on}$, of $0.44 \pm 0.01\ \mu M^{-1}\ s^{-1}$, and a slow conformational selection step where the binding competent, closed conformation is repopulated ($k_{close} = 2.9 \pm 0.1\ s^{-1}$) (Fig. 6b). SHP099 dissociation is slow for FL-WT ($k_{off} = 0.029 \pm 0.001\ s^{-1}$, Fig. 6c), one characteristic of a good therapeutic. The only rate constant that could not be directly measured is $k_{open}$, the opening rate. Fortunately, this rate can be calculated from the open/closed equilibrium constant determined by NMR and the closing rate obtained from the stopped-flow experiment. Using all intrinsic rate constants shown for the conformational selection binding scheme (Fig. 6f), an apparent dissociation constant of $0.074 \pm 0.003\ \mu M$ for SHP099 is calculated (using Eq. (4)), in

excellent agreement with the $K_D$ determined by ITC ($0.073 \pm 0.058\ \mu M$, Fig. 5a).

The kinetic traces for FL-E76K with SHP099 are single exponential and the observed rate constant increases linearly with drug concentration, indicative of the binding step (Fig. 6d). Two major differences are apparent in the binding kinetics for FL-E76K with SHP099 compared with wild-type. First, we observe an order of magnitude slower rate of binding ($k_{obs}$, Fig. 6d), which is mainly due to the large decrease in population of the binding competent, closed conformation (Fig. 6f). Note that $k_{obs}$ is the product of the true on-rate ($k_{on}$) and the population of the closed state (Eq. 2). The reduced $k_{obs}$ serves as another validation for the binding scheme (Fig. 6f). Second, SHP099 dissociates faster from the mutant protein ($k_{off} = 0.10 \pm 0.01\ s^{-1}$, Fig. 6e). The conformational selection step could not be detected here likely due to the apparent rate being almost identical to the $k_{obs}$ for binding. The overall $K_D$ calculated from the corresponding microscopic rate constants ($1.5 \pm 0.9\ \mu M$, Fig. 6f) is in excellent agreement with the experimentally measured $K_D$ from ITC ($1.5 \pm 0.2\ \mu M$, Fig. 5a).

Having determined the population thermodynamics and kinetics for opening/closing of SHP2 both in wild-type and mutant forms, it is now possible to analyze the enzymatic data for FL-WT and FL-E76D (Supplementary Fig. 7c–e). The observed activities result from a complex relationship between turnover in the fully open conformation and its interconversion with the closed state. The former we understand from the activity of the PTP domain (Fig. 3c), whereas the latter is now determined from NMR and stopped-flow data. Combining these elements, the expected activity for FL-WT and FL-E76D can be simulated. Remarkably, a robust correlation is observed between the calculated and experimental turnover rates (Supplementary Fig. 7c–e).

The stopped-flow data showed that the interconversion between the open and closed states in FL-WT is slow ($k_{close} + k_{open} \sim 3.3\ s^{-1}$). This implies that residues with sufficiently large chemical shift differences between these states could be in the slow NMR exchange regime. To test this hypothesis, a long [$^1$H-$^{15}$N]-TROSY-HSQC spectrum of perdeuterated FL-WT was performed. Strikingly, additional peaks were found for N-SH2 resonances along domain interfaces (Supplementary Fig. 8). The position of these minor peaks for N-SH2 residues were identical to their respective residues found in the tandem-SH2 construct. Additionally, a qualitative volumetric comparison of minor and major peaks found the minor state was populated within error to the open percentage calculated prior from the linear chemical shift changes in the PTP domain.

The comparable open percentages found from minor peaks in the N-SH2/C-SH2 and the fast-exchanging residues in the PTP domain implicates a second process occurring in SHP2 connected with opening/closing. As mentioned prior, the movement of the WPD loop in the active site represents a crucial dynamical process in catalysis. Although the open form of SHP2 allows for the WPD loop to convert between in and out states, the closed form quenches this interconversion and pushes the WPD loop in the out conformation. Based upon elegant work for the SHP2 homolog PTP1B, which found that the WPD loop exchanges fast between the in and out conformation[22,25,36], we conjecture that for residues in the PTP domain chemical shifts are partitioned between a closed, out state and an open, in/out state. Therefore, the chemical shifts of PTP residues change linearly with the conformational equilibrium of the slow open/closing process. Although additional experiments would be required to prove our hypothesis, the following lines of evidence provide a robust independent verification for the determined open/closed populations: (i) the comparable populations seen using either the minor

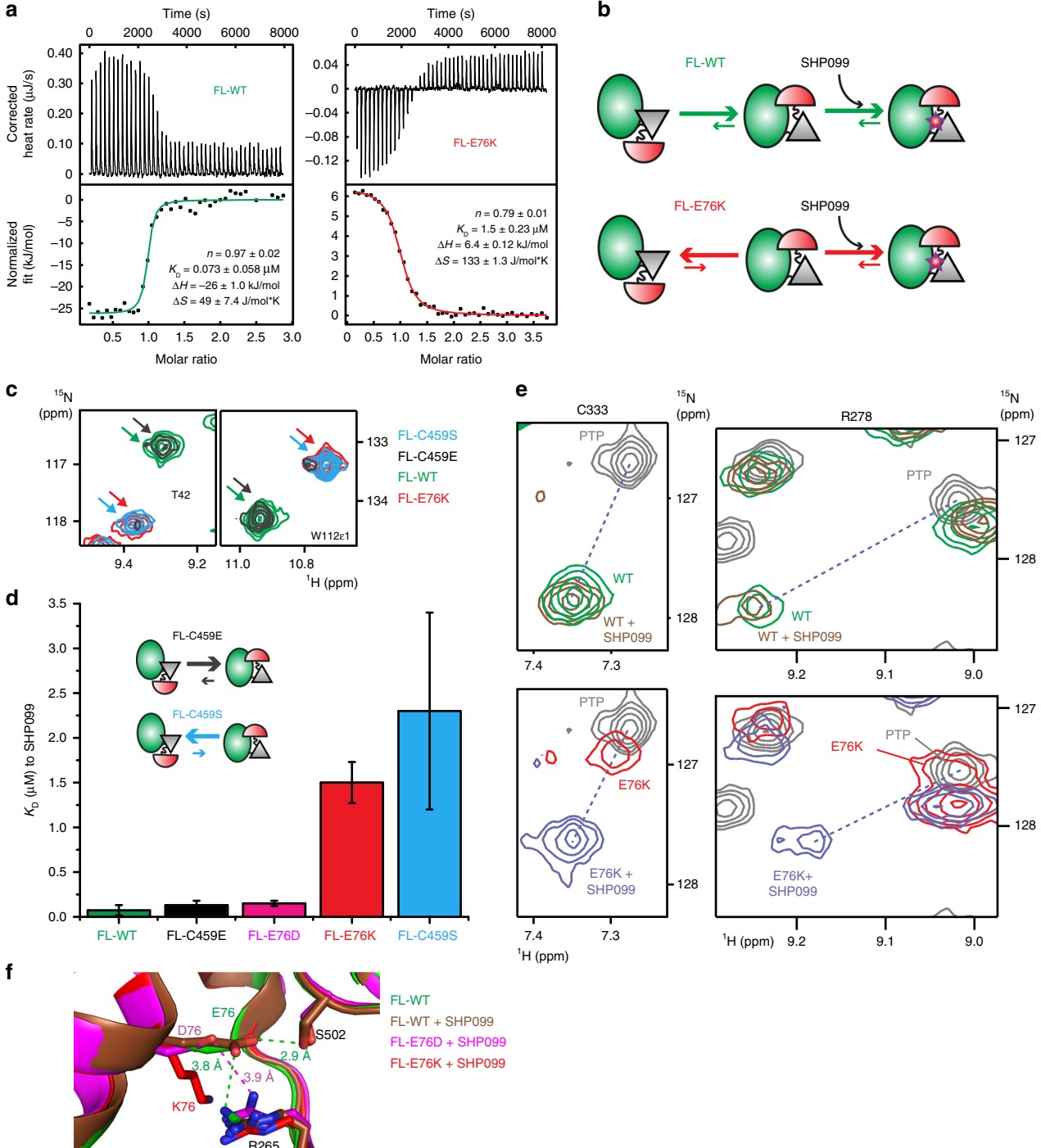

**Fig. 5** Comparing thermodynamics of SHP099 binding to wild-type and mutants of SHP2. **a** ITC profiles for the binding of SHP099 to FL-WT and FL-E76K (left, 30.5 μM of FL-WT titrated with 320 μM of SHP099; right, 82 μM of FL-E76K titrated with 800 μM of SHP099). **b** Scheme representing the proposed SHP099 binding mechanism with inverted equilibria between open and closed conformations as illustrated by size of kinetic arrows. **c** Selected regions of NMR spectra show that FL-C459E and FL-C459S resemble FL-WT and FL-E76K, respectively. **d** Bar graph summarizing the SHP099 dissociation constants determined by ITC for the denoted proteins (see also Supplementary Fig. 6e-f) together with a scheme for the open/close equilibrium of the dead mutants; uncertainties were derived from the fit to the data. **e** Chemical shift analysis to determine the population of the closed forms of FL-WT and FL-E76K. Dashed lines indicate the linear path between peaks in the PTP and FL-WT + SHP099 spectra. Using PTP and SHP099-bound chemical shifts for the completely open and closed states, respectively, the fraction of the closed state for FL-WT and FL-E76K are 0.90 and 0.04, respectively. **f** Superimposition of FL-WT (green), and protein/drug complexes FL-WT (brown), FL-E76D (magenta) and FL-E76K bound to SHP099 (red), reveals that the E76K mutation disrupts a hydrogen bond (E76–S502) and a salt bridge (E76–R265) between the N-SH2 and PTP domains, whereas the E76D mutation maintains the electrostatic interaction between D76 and R265

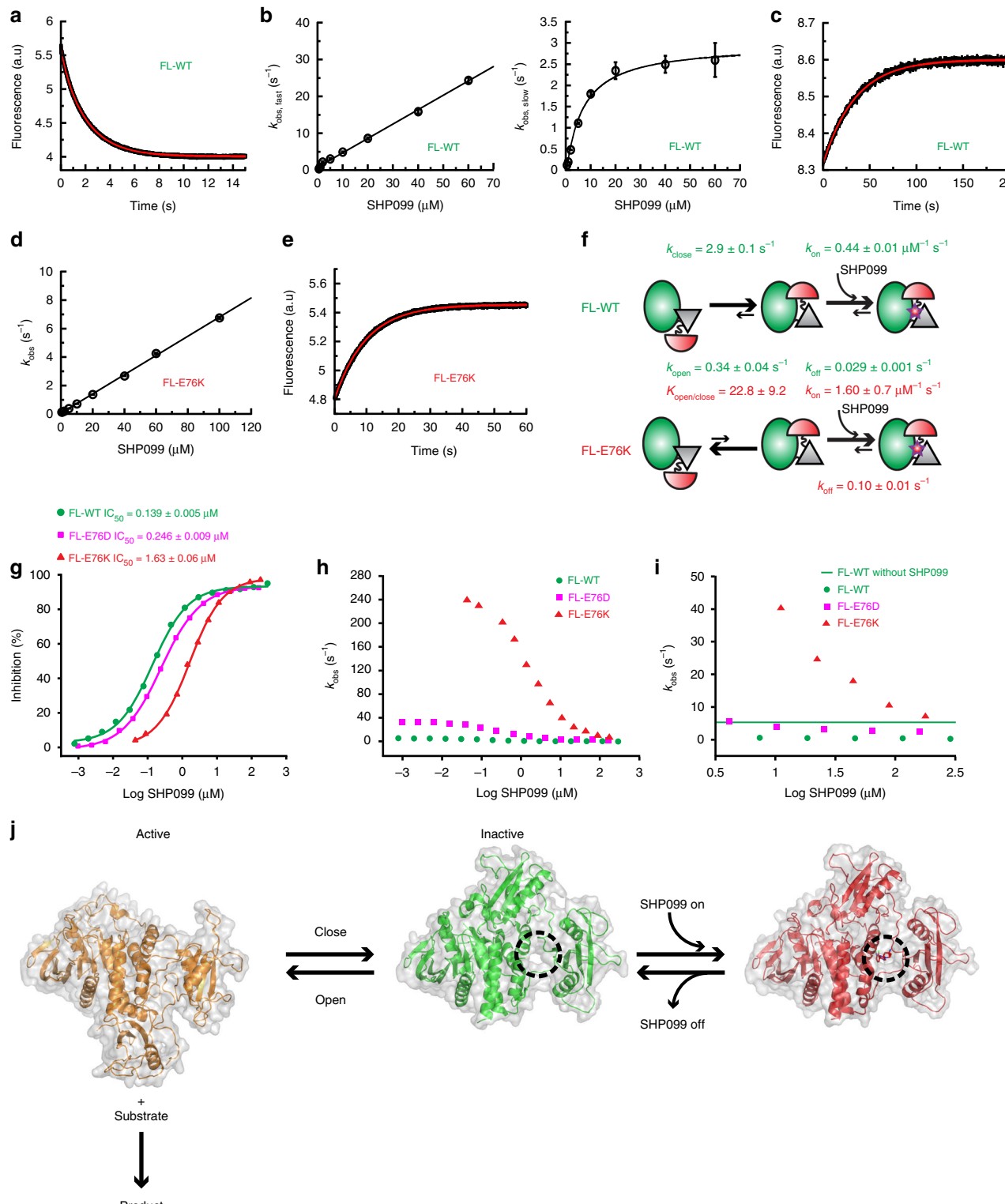

peaks or linear shifts in the PTP of FL-WT, (ii) the remarkable similarity between calculated/observed activities, and (iii) the agreement between calculated/observed affinities for SHP099.

**Implications for drug efficacy.** Stopped-flow kinetics of SHP099 with both wild-type and mutant SHP2 found that reduced affinity in FL-E76K arises from an inverted open/closed population, as well as an expedited $k_{off}$. Having determined the binding mechanisms of SHP099 for both WT and mutant SHP2, our

results beg the question: What are the implications of these kinetic differences for drug development? Although SHP099 has been shown to possess excellent potency against FL-WT[1], it should have a much weaker IC$_{50}$ for the oncogenic mutants based on our binding kinetics. Indeed, the FL-E76K mutant experiences a significantly weakened inhibition profile for SHP099 with a >10-fold higher IC$_{50}$ (Fig. 6g). As predicted from a conformational selection model, FL-E76D has an intermediary IC$_{50}$ value as it samples a closed population in between WT and E76K.

**Fig. 6** Binding kinetics of SHP099 explains its reduced potency for FL-E76K. **a** Representative stopped-flow fluorescence trace shows a double-exponential decay after mixing 0.5 μM FL-WT and 2.0 μM SHP099. **b** The extracted rates for the double-exponential decays are plotted against the SHP099 concentration. The rates for the faster phase are linearly dependent on the SHP099 concentration and represent the binding step (left), whereas those for the slower phase plateau and reflect the conformational selection step (right). **c** Dissociation kinetics measured after an 11-fold dilution of the FL-WT/ SHP099 complex. **d** Binding kinetics of FL-E76K appears single exponential with a decrease of about sevenfold in the observed rate constants compared with FL-WT in **b**. **e** Dissociation kinetics of the FL-E76K/SHP099 complex measured as described in **c**. **f** The measured and calculated rates are shown in the proposed binding scheme. The $k_{open}$ is calculated from the experimentally measured $k_{close}$ and the NMR-derived equilibrium constant. **g** Inhibition constants (IC$_{50}$) and **h**, **i** plot of observed activity as function of SHP099. **i** Is a zoom-in of **h** and the green line indicates basal wild-type activity without drug. Uncertainties in **g** were obtained from the fit to the data. **j** Scheme of conformational sampling and its role in catalysis and drug binding. Surface representation of ΔN-SHP-2 (gold, PDB 6cmq) as the open/active conformation, ligand-free FL-WT (PDB 4dgp[15]) showing a cavity formed between the two interdomain linkers and the PTP domain where SHP099 binds (black dotted circle) and FL-E76K bound to SHP099 (red, PDB 6cms). The experiments in **a**, **c**, and **e** were repeated five times. Data points and error bars in **b** and **d** are the average and standard deviations of the analysis of replicate experiments ($n = 5$). In **f**, the values and standard deviations were obtained from the fitting

Although IC$_{50}$ values are generally reported and used as a measure for the effectiveness of a drug, activating mutants require more than simply half inhibition to re-normalize cellular activity to its basal level. The E76D and E76K mutations increase the basal activity of FL-WT from 5 s$^{-1}$ to 34 s$^{-1}$ and 250 s$^{-1}$ at 500 μM substrate, respectively. By plotting the actual $k_{obs}$ for phosphatase activity instead of percentage inhibition, it is apparent that almost 4.1 μM and >180 μM of SHP099 is needed to properly inhibit FL-E76D and FL-E76K, respectively, and restore WT activity levels (Fig. 6h, i).

Finally, the reduced affinity for SHP099 to mutant forms becomes exacerbated under activating circumstances. In the presence of a synthetic, bisphosphorylated tyrosine-activating peptide, 2P-IRS-1[1], the free energy of the open conformation becomes stabilized with respect to the closed state[10]. Therefore, the change in activity upon SHP099 titration is a result of a complex coupling of several equilibria, in which peptide binding and drug-binding shift the closed/open equilibrium in the opposite directions. As SHP099 inhibits SHP2 by binding to the closed form, it is evident that the apparent IC$_{50}$ values must be higher in the presence of the activating peptide than in its absence. Consequently, all forms of SHP2 have 6- to 12-fold increases in IC$_{50}$ (Supplementary Fig. 9).

## Discussion

The phosphorylation writers, protein kinases, have long been the traditional target of chemotherapeutic drug development. In contrast, the development of orthosteric inhibitors for the erasers, protein phosphatases, have been stymied by the phosphatases' overtly promiscuous active sites, with shallow, electrostatically driven substrate-binding cavities. Therefore, the recent development of an allosteric inhibitor, SHP099, provides an elegant opportunity for targeting these undruggable enzymes. In addition, allosteric inhibitors likely offer the advantage of increased selectivity since extant phosphatases have evolved differential allosteric regulation while the active sites remained conserved. However, just as single point mutations can dramatically alter the phenotype of a cell, we show here that it can transform the free-energy landscape of an enzyme. With this in mind, it is crucial to evaluate the potency of SHP099 in its native mutant landscape and characterize the molecular mechanism of inhibition for future development of improved inhibitors for cancer-driving mutant proteins.

While often overlooked, it is fundamental to understand the molecular mechanism underlying cancer mutations. The large shift in population toward the open, active SHP2 conformation by the single point mutant E76K, together with an atomic resolution structure of the open conformation, sheds light into the activation mechanism (Fig. 6j). Opening releases, the steric hindrance for substrate access, but also allows WPD loop to sample the active WPD-in conformation. In support of our findings, recent NMR CPMG relaxation dispersion experiments on the PTP domain of PTP1B, a SHP2 paralog, as well as room-temperature crystallography, have shown interconversion between WPD-in and -out conformations serves as a fundamental dynamic barrier for catalysis[22,25].

The characterization of the free-energy landscape changes for SHP099 binding to wild-type and mutant forms (both activating mutations and catalytically dead mutants) may deliver guidance for the development of new inhibitors with improved potency against E76K and other cancer-associated mutations. Most importantly, our findings advocate for more rigorous experiments than only IC$_{50}$ measurements on activating mutant proteins. For example, in the case of E76K activity must be reduced by 98% to assure the same basal-level activity as for wild-type SHP2, which falls at the far tail of the inhibition curve. Furthermore, it is the signaling-off state and not the peptide-bound, activated state of the phosphatase that is altered by the cancer-driving mutations studied here. Because E76K is already fully active, binding of a phosphorylated peptide does not induce further activation. Therefore, we primarily focused here on characterizing the inhibition of SHP2 in the absence of the activating peptide.

Dual inhibition might be a promising direction forward for the SHP2 mutant proteins given the huge reduction in catalytic turnover that is needed for E76K and E76D, and the small percentage of the binding competent, closed state. This approach has recently been investigated by Fodor et al. for SHP2[37] and was successfully developed for Bcr–Abl[38]. A better understanding of the two interconverting conformations and the dynamics of this equilibrium as characterized here could guide to build in positive cooperativity for dual inhibition, and double drugging may prevent the emergence of resistance mutations.

Lessons learned for SHP2 are likely to have broader implications since many phosphatases and kinases are regulated by similar principles of SH2- or SH3-domain binding to the catalytic domain. As regulation has likely evolved to be specific, allosteric inhibitors open opportunities for highly selective, novel drugs.

## Methods

**Preparation of recombinant proteins**. The optimized DNA sequence encoding human FL-WT lacking the C-terminal tail (residues 1–529, UniProt: Q06124-2) (Supplementary Table 1) and ΔN-SH2 (residues 104–529) (Supplementary Table 2) was synthesized with an N-terminal Tobacco Etch Virus protease (TEV)-cleavage site, and sub-cloned into pET-28a(+) expression vector containing a His-tag on the N-terminus by Genscript. Mutations in SHP2 were introduced using QuikChange Lightning mutagenesis kit (Agilent Technologies) with primers from IDT (Supplementary Table 3). Tandem N-SH2 and C-SH2 (tandem-SH2, residues 1–217) and N-SH2 (residues 1–106) constructs were generated by introducing stop

codons in the FL-WT expression plasmid. C-SH2 (residues 104–217) was constructed by introducing a stop codon in the ΔN-SH2 expression plasmid. To prepare the PTP construct (residues 216–529), a gBlocks gene fragment containing the encoding sequence in addition to a C-terminal TEV-cleavage site was synthesized by IDT and inserted into pLATE31 expression vector containing a His-tag on the C-terminus using aLICator Ligation Independent Cloning and Expression System (Thermo Fisher Scientific). All fusion proteins were expressed in *Escherichia coli* BL21(DE3) competent cells (NEB) and purified by affinity chromatography on a HisTrap column (GE Healthcare). The His-tag was subsequently cleaved off with in-house made His-tagged TEV protease overnight at 4 °C while dialyzing against HisTrap binding buffer (50 mM Tris pH 8, 500 mM NaCl, and 1 mM tris(2-carboxyethyl) phosphine (TCEP)). Cleaved protein was separated from His-tag and TEV protease by HisTrap purification. Flow through was collected and further purified by size exclusion chromatography on a HiLoad 16/600 Superdex 75 pg column (GE Healthcare). All recombinant SHP2 constructs except for PTP contained three extraneous amino acids (GSG) preceding their natural sequences. The PTP sequence was preceded by one Met and followed by six amino acids from the TEV-cleavage site (ENLYFQ).

**Steady-state kinetics**. Phosphatase activity was measured by monitoring the dephosphorylation of the synthetic substrate DiFMUP (6,8-difluoro-4-methyl-lumbelliferyl phosphate) to the product DiFMU (Thermo Fisher Scientific). The reaction is carried out in activity buffer (50 mM Bis-Tris pH 6.5, 50 mM NaCl, 1 mM TCEP, 0.05% Tween 20, 0.3 mg mL$^{-1}$ bovine serum albumin) at 35 °C and was started by adding 10 μL of enzyme to 90 μL of substrate in a 96-well plate (Corning—Ref 3994). The absorbance at 366 nm was continuously monitored in a SpectraMax i3x plate reader (Molecular Devices) and converted to product concentration using a calibration curve obtained from a serial dilution of the pure product. Final enzyme concentrations used were 10–20 nM for FL-WT and 50–100 pM for the other constructs. Uncertainties were calculated from the standard deviation of the observations ($n = 3$).

Initial reaction rates plotted versus substrate concentration differs from a Michaelis–Menten profile (see Fig. 3c and Supplementary Fig. 7a). The $k_{cat}$ of the fully active SHP2 constructs did not remain constant at saturating substrate concentrations, but instead gradually dropped and leveled off at a second observed rate. The drop in $k_{cat}$ indicates partial or full substrate inhibition. The data were fitted to the simplest two site sequential substrate-binding model (Supplementary Fig. 10a) described analytically by Eq. (1)[39] and fitted in OriginPro 2018 (OriginLab, Northampton, MA). A modified form of the Adair–Pauling model was used to account for two distinct substrate-binding sites[39]. This model predicts that the enzyme can form two active enzyme/substrate complexes (ES and ESS) each one exhibiting different turnover rates ($k_{cat,1}$ and $k_{cat,2}$) and dissociation constants ($K_{D,1}$ and $K_{D,2}$). This model was chosen because the reaction rates reached a second $k_{cat}$ value instead of dropping to zero, which would be indicative of formation of a dead-end complex and full substrate inhibition.

$$k_{obs} = \frac{k_{cat,1}\frac{[S]}{K_{D,1}} + k_{cat,2}\frac{[S]^2}{K_{D,1}K_{D,2}}}{1 + \frac{[S]}{K_{D,1}} + \frac{[S]^2}{K_{D,1}K_{D,2}}} \qquad (1)$$

The steady-state simulations for FL-WT and FL-E76D (Supplementary Fig. 7) were performed with KinTek Explorer[40,41] using the scheme in Supplementary Fig. 10b. Rate constants derived from the experiments and used in the simulations are: (i) the equilibrium constant ($K_{close/open}$) of opening/closing from the NMR data, (ii) the dissociation constants ($K_D$) and $k_{cat}$ values from the steady-state kinetics data of PTP measured at 35 °C, and (iii) the $k_{close}$ that was estimated to be 5.8 s$^{-1}$ (i.e., roughly twofold higher than the value obtained from stopped-flow experiments at 25 °C). The purpose of these simulations was to qualitatively explain the $k_{obs}$ for the DiFMUP activity data with FL-WT and FL-E76D. Using the known kinetic parameters (see above), we are able to recapitulate the activity data and show that the difference can be explained solely by the difference in populations. The best-fit values in the simulations for the open, active conformation (11 and 30% for FL-WT and FL-E76D, respectively) agree well with the experimentally determined values.

**Inhibition assay**. Phosphatase activity was measured in the presence of various concentrations of the inhibitor SHP099 (MedChemExpress) in activity buffer containing 500 μM DiFMUP in absence and presence of 5 μM synthetic, biphosphorylated IRS-1 peptide (H$_2$N-LN(pY)IDLDLV(dPEG8)LST(pY)ASINFQK-amide, Anaspec). Percentage of inhibition was plotted against the logarithmic concentration of SHP099 and the IC$_{50}$ values and uncertainties were obtained after fitting the data to the Boltzmann function in OriginPro 2018 (OriginLab, Northampton, MA).

**ITC**. Protein samples were dialyzed against HEPES buffer (25 mM HEPES pH 7.5, 100 mM NaCl, 2 mM TCEP, and 1% dimethyl sulfoxide (DMSO)), and SHP099 powder was dissolved in the same buffer. ITC experiments were

carried out in a Nano ITC instrument (TA Instruments) at 25 °C. The titrations were performed by injecting 1.0 μL aliquots of SHP099 into the calorimeter cell containing a 170 μL solution of SHP2 (30.5 μM FL-WT and 300 μM SHP099; 82 μM FL-E76K, and 800 μM SHP099; 25.3 μM FL-E76D and 300 μM SHP099; 35.2 μM FL-C459E and 300 μM SHP099; 38.6 μM FL-C459S and 300 μM SHP099) with a constant stirring speed at 150 rpm. The data were analyzed with the NanoAnalyze using the independent fit model. All the uncertainties were estimated by the native Statistics module with 1000 synthetic trials and 95% confidence level.

**Stopped-flow experiment**. Stopped-flow experiments were performed with the Applied Photophysics SX-20 instrument equipped with a temperature control unit set at 25 °C. Changes in the intrinsic tryptophan fluorescence of SHP2 accompanying the binding and dissociation of SHP099 were monitored, using an excitation wavelength of 295 nm (2.3 nm bandwidth) and a long-pass 320 nm cutoff filter to detect emission. The flow system was rinsed with degassed buffer comprised of 50 mM Bis-Tris pH 6.5, 50 mM NaCl, 2 mM TCEP, and 0.5% DMSO, to minimize photobleaching. The proteins and drug were in the same buffer and degassed with ThermoVac (MicroCal) at 25 °C prior to experiments. Binding experiment was initiated by rapidly mixing 1 volume of 5.0 μM protein with 10 volumes of varying concentrations of SHP099 (0.5–60 μM for FL-WT and 0.5–100 μM for FL-E76K). To initiate dissociation, 1 volume of the mixture containing protein and SHP099 was pre-incubated for 1 h and mixed with 10 volumes of buffer. For FL-WT, the mixture contained 0.6 μM of protein and 0.6 μM of SHP099. For FL-E76K, it contained 5.0 μM of protein and 5.0 μM of SHP099. For each experiment, at least five replicate measurements were performed and the traces were averaged.

The fluorescence traces of the binding experiments for FL-WT were fitted to a double-exponential function by nonlinear regression using functions built into KinTek Explorer[40,41] or Pro-Data viewer (Applied Photophysics Ltd). The rates for the faster decaying phase exhibited a linear dependency on the SHP099 concentration, thus the slope represents the observed $k_{on}$ of SHP099 to FL-WT. The true $k_{on}$ can be extracted from the slope of the linear dependency using $K_{open/close}$ (Eq. (2)) that was calculated using the populations determined from the NMR chemical shift analysis described later.

$$k_{obs} = k_{on} * \left(\frac{1}{K_{open/close} + 1}\right) * [SHP099] + k_{off} \qquad (2)$$

The rates for the slower decaying phase exhibited a hyperbolic shape and were fitted to Eq. (3) using KaleidaGraph version 4.5.3 (Synergy) to yield the $k_{close}$ of the conformational selection step.

$$k_{obs}(slow) = \frac{k_{close}[SHP099] + \frac{(k_{close}+k_{open})}{k_{on}}k_{off}}{\frac{(k_{close}+k_{open})}{k_{on}} + [SHP099]} \qquad (3)$$

where, $k_{open}$, $k_{close}$, $k_{on}$, and $k_{off}$ are defined in Fig. 6f.

The binding traces for FL-E76K exhibited single-exponential decays and the rates were linearly dependent on the SHP099 concentration. The true $k_{on}$ rate can be calculated using Eq. (2).

The overall dissociation constants were calculated from the intrinsic rate constants based on the mechanism of conformational selection followed by inhibitor binding (Eq. (4)) (Supplementary Fig. 10c).

$$K_D^{obs} = (K_{open/close} + 1) * K_D \qquad (4)$$

where $K_{open/close} = \frac{k_{open}}{k_{close}}$ and $K_D = \frac{k_{off}}{k_{on}}$.

The uncertainties in the calculated dissociation constant using the equation above are obtained using standard error propagation.

**X-ray crystallography**. All versions of SHP2 were crystallized at 291 K by sitting drop vapor diffusion using a Cryschem M plate (Hampton Research) with 500 μL of crystallization solution in the reservoir with drops consisting of 1 μL protein and 1 μL reservoir solution. Proteins were dialyzed in buffer (20 mM Tris pH 8.0, 150 mM NaCl and 2 mM TCEP) prior to crystallization experiment.

For FL-E76K, a 10 mg mL$^{-1}$ protein solution containing 2 mM SHP099 was mixed with 100 mM bicine pH 9, 100 mM NaCl and 19% (w/v) PEG MME 550. Crystals were bathed in cryoprotectant (100 mM bicine pH 9, 100 mM NaCl, 35% (w/v) PEG MME 550 and 2 mM SHP099) and flash cooled in liquid nitrogen. For FL-E76D, crystals were grown by mixing 8.5 mg mL$^{-1}$ protein with 1.5 mM SHP099 in 15% (w/v) PEG 20,000 and 10 mM potassium hydrogen tartrate and cryoprotected with Paratone N. For ΔN-SH2, crystals were obtained by mixing 15 mg mL$^{-1}$ protein with 100 mM ammonium formate, 22% (w/v) PEG 3350 and 1.5% (w/v) xylitol. Cryoprotectant solution consisted of mixing 2 μL of reservoir solution with 1 μL of 70% (w/v) xylitol. For FL-C459E, crystals were obtained by mixing 10 mg mL$^{-1}$ protein in 200 mM lithium nitrate, 20% (w/v) PEG 3350 and cryoprotected with Paratone N.

Data collection was carried out in ALS beamlines 8.2.1 and 8.2.2 and SSRL beamline 14-1 as summarized in Supplementary Table 4. Diffraction images were indexed and integrated in iMOSFLM[42], scaled and merged in AIMLESS[43], the structures were solved by molecular replacement using PHASER[44] searching for the full structure (FL-E76K, -E76D and -C459E) or individual domains (ΔN-SH2) of the FL-WT structure previously published (PDB 4dgp)[15]. Model building and refinement were conducted in Coot[45] and PHENIX[46], respectively. In addition, two data sets for FL-E76K were collected and merged using BLEND[47] to increase completeness at higher resolution. Maximum resolution limit for ΔN-SH2 was determined based on the $CC_{1/2}$ (>0.3) along the $k$ axis due to anisotropic diffraction. Initial analyses in Xtriage[46] revealed ΔN-SH2 structure contained four molecules in the asymmetric unit (ASU) and FL-C459 2 molecules in ASU with a pseudo translational symmetry. The assigned space groups were validated in ZANUDA[48], using the structures solved in the P1 space group as input. The position of ASU in the cell was standardized using ACHESYM[49] before deposition to the Protein Data Bank (FL-C459E: 6cmp; ΔN-SH2: 6cmq; FL-E76D + SHP099: 6cmr; FL-E76K + SHP099: 6cms).

Protein structure images were rendered in Chimera[50] and PyMOL[51]. Rigid body domain rotations were evaluated using RotationAxis (draw_rotation_axis.py; python script by Pablo Guardado Calvo and available from PyMOLWiki, https://pymolwiki.org).

### SAXS data collection and prediction of SHP2 open state.
SAXS profiles were acquired in SSRL beamline BL4-2. FL-WT and FL-E76K samples were buffer exchanged to 50 mM ADA pH 6.5, 2 mM TCEP using gel filtration and concentrated to 14.2 mg mL$^{-1}$ and 12.2 mg mL$^{-1}$, respectively, using a 10 kDa cutoff Vivaspin centrifugal concentrator. The flow-through buffer was kept for background scattering subtraction and sample dilution. A concentration series of each protein was used for scattering experiments (FL-WT 14.2, 7.1, 3.55, 1.775, 0.8875 mg mL$^{-1}$, FL-E76K 12.2, 6.1, 3.05, 1.525, 0.7625 mg mL$^{-1}$). Ten images per sample were acquired using 1-s exposure to an 11 keV beam with a Rayonix MX225-HE detector positioned 1.65 m from the sample. Data processing was performed using the SAXSPipe software, available in the beamline, and the ATSAS[52] package. The final profile for each protein was obtained after merging the data from the concentration series using SAXS Merge[53]. SAXS envelopes were calculated using DAMMIN[54].

MODELLER 9.20[55] was used to obtain a homology model of SHP2-E76K using the crystal structures of SHP1 in the open conformation (PDB 3ps5)[21] and the ΔN-SH2 SHP2 as templates. The missing loops from the crystal structure were modeled to reflect the native structure in solution and the model stereochemistry was optimized by the Geometry Minimization tool available in PHENIX[46]. The N-SH2 was then treated as a rigid body connected by a flexible linker (residues 99–111) and allowed to sample 10,000 conformations using the rapidly exploring random trees algorithm in IMP[56]. The SAXS profile for each conformation was calculated using FOXS[57] and compared with the FL-E76K profile in MultiFoXS[58]. The model with the best score was then fed to AllosMod[59] for a final loop refinement step also using the FL-E76K SAXS profile as target. The best model was deposited in Model Archive[60,61] under accession code ma-aks8e and together with the raw SAXS data also deposited in SASBDB[62] (FL-WT: SASDEN4; FL-E76K: SASDEP4). SAXS data collection and processing parameters are summarized in the Supplementary Table 5[63].

### NMR spectroscopy.
Samples for NMR spectroscopy contained 0.2–0.9 mM of [$^{15}$N]-, [$^{13}$C, $^{15}$N]-, or [$^{2}$H, $^{13}$C, $^{15}$N]-labeled proteins in NMR buffer (50 mM ADA pH 6.5 and 2 mM TCEP), supplemented with 10% D$_2$O and 0.02% NaN$_3$. The [$^{2}$H, $^{13}$C, $^{15}$N]-labeled FL-WT and PTP proteins were expressed in M9 minimal medium containing 2.0 g L$^{-1}$ of U-[$^{2}$H, $^{13}$C]-glucose and 1.0 g L$^{-1}$ of $^{15}$NH$_4$Cl in 99.8% D$_2$O (Cambridge Isotope Laboratories, Tewksbury, MA, USA). All other [$^{15}$N]- and [$^{13}$C, $^{15}$N]-labeled samples were expressed in H$_2$O-based M9 minimal medium containing 1.0 g L$^{-1}$ of $^{15}$NH$_4$Cl and 2.0 g L$^{-1}$ the appropriate carbon source (U-[$^{13}$C]-glucose or D-glucose for [$^{13}$C, $^{15}$N]-labeled and $^{15}$N-labeled proteins, respectively). NMR data were acquired at 308 K, unless stated otherwise, on an Agilent DD2 600 MHz and a Bruker Avance II 800 MHz spectrometer, each equipped with triple-resonance cryoprobes. All of the NMR spectra were processed with NMRPipe[64] and analyzed with CcpNmr Analysis version 2.4.2[65]. Backbone chemical shift assignments (H$^N$, N, C$^\alpha$, C$^\beta$) for [$^{2}$H, $^{13}$C, $^{15}$N]-labeled FL-WT and PTP samples containing ~0.2 mM protein, were based on TROSY-versions of three-dimensional (3D) HNCA, constant-time (CT) HNCACB and HNCO spectra that are part of the Bruker pulse sequence library[66,67]. An NMR titration experiment with the unlabeled N-SH2 being added into $^{15}$N-labeled PTP, as well as a TROSY-version of the 3D $^{15}$N-edited $^{1}$H, $^{1}$H-NOESY experiment[68] (mixing time 120 ms) on [$^{2}$H, $^{13}$C, $^{15}$N]-labeled FL-WT were conducted to verify the assignments for FL-WT. The backbone chemical shift (H$^n$, N, C$^\alpha$, and C$^\beta$) assignments in [$^{13}$C, $^{15}$N]-labeled tandem-SH2-E76K (sample containing ~0.9 mM protein) was based on 3D HNCA and HNCACB spectra that are part of the Agilent BioPack pulse sequence library[69]. The $^{1}$H and $^{15}$N chemical shifts in $^{15}$N-labeled tandem-SH2-WT were assigned based on the similarity to the 2D [$^{1}$H-$^{15}$N]-TROSY-HSQC spectra[70,71] of tandem-SH2-E76K. The assignment of the side chain indole signals of W6 and W112 were based on their unique peak positions in tandem-SH2-E76K or -WT, and their similarity to the same peaks in N-SH2 and C-SH2, respectively, which both have only one tryptophan residue.

Earlier NMR experiments on only the C-SH2 domain[72] or a variety of constructs[73] were performed in phosphate buffer, which we deemed problematic as the phosphate is likely to interact with the protein as its function is to recognize phosphorylated substrates. Nevertheless, we used their data in conjunction with an NMR titration experiment of [$^{1}$H, $^{15}$N]-labeled tandem-SH2 with inorganic phosphate to compare both data sets and substantiate our assignments. Due to the lower thermal stability than FL-WT, all 3D spectra on PTP and tandem-SH2-E76K were acquired at 298 K, and the 2D [$^{1}$H-$^{15}$N]-TROSY-HSQC spectra at 308 K of both proteins were assigned by following the peaks in a series of 2D spectra recorded at 298, 303, and 308 K. Cross peaks in 2D [$^{1}$H-$^{15}$N]-TROSY-HSQC spectra of $^{15}$N-labeled SHP2 mutants were assigned based on similarity to the TROSY spectra of FL-WT, tandem-SH2-E76K, or PTP. The chemical shift assignments of FL-WT, PTP, and tandem-SH2-E76K have been deposited in the BioMagResBank[74] with accession codes, 27520, 27521, and 27522, respectively.

To investigate binding of SHP099 to the SHP2 constructs, a concentrated SHP099 sample (10 mM in NMR sample buffer in addition of 10% DMSO) was added in steps to the $^{15}$N-labeled, ligand-free protein (340 μM and 320 μM for FL-WT and FL-E76K, respectively) to obtain final inhibitor concentrations of 0, 80, 160, 350, and 640 μM, and [$^{1}$H-$^{15}$N]-TROSY-HSQC spectra were acquired. A spectrum of the $^{15}$N-labeled ligand-free protein in presence of 1% of DMSO was acquired as a control and no change of peak position was observed. The chemical shift differences are calculated by CcpNmr Analysis using the equation below:

$$\Delta\delta(\text{ppm}) = \sqrt{(\Delta\delta_H)^2 + (0.15 * \Delta\delta_N)^2} \qquad (5)$$

### Determination of populations by NMR chemical shifts.
For residues in the PTP domain of SHP2, chemical shifts changed linearly with the population of open and closed conformations. Using the inhibitor-bound complex as a purely, closed species and the PTP domain as purely, open species, the open and closed populations of FL-WT, FL-E76D, and FL-E76K were determined through chemical shift projection analysis[75] of cross peaks in [$^{1}$H-$^{15}$N]-TROSY-HSQC spectra. All data sets contained (128, 1024) complex points in the ($^{15}$N, $^{1}$H) dimensions with acquisition times of (43.8, 91.8 ms). The $^{15}$N time domain was doubled using forward-backward linear prediction before apodization with a cosine-squared window function, and subsequent Fourier transformation. The resultant chemical shifts were used to calculate the fraction of the closed state from the following equations:

$$\vec{A} = \begin{bmatrix} \delta H - \delta H_{\text{closed}} \\ 0.15 \times (\delta N - \delta N_{\text{closed}}) \end{bmatrix}$$

$$\vec{B} = \begin{bmatrix} \delta H_{\text{closed}} - \delta H_{\text{open}} \\ 0.15 \times \left(\delta N_{\text{closed}} - \delta N_{\text{open}}\right) \end{bmatrix}$$

$$\cos(\theta) = \frac{\vec{A}\cdot\vec{B}}{|\vec{A}| * |\vec{B}|}$$

$$F = \frac{|\vec{A}| * \cos(\theta)}{|\vec{B}|} \qquad (6)$$

where $\delta H$ and $\delta N$ are the chemical shift values taken from the [$^{1}$H-$^{15}$N]-TROSY-HSQC spectra of FL-WT, FL-E76D, and FL-E76K. $\delta H_{\text{open}}$ and $\delta N_{\text{open}}$ are represented by the chemical shift values in PTP, and $\delta H_{\text{close}}$ and $\delta N_{\text{close}}$ are given by the chemical shift values in SHP099-inhibited FL-WT and FL-E76K, respectively. Although many residues exhibited changes with respect to inhibitor-bound and PTP spectra, only residues possessing a linear chemical shift pattern were analyzed (±10°). Additionally, residues with $|B| < 0.03$ ppm, $0 > F$, and $1 < F$ were removed from further analysis. Finally, residues within 15 Å of the SHP099 binding site were not considered.

Once the set of residues had been reduced, the open/closed percentages were determined through a maximum likelihood estimation approach. Assuming the observed chemical shifts are derived from a Gaussian distribution around the true chemical shift for each residue, Eq. 6 was used to construct a log likelihood function for our data[76]:

$$l(F, \sigma) = \sum_{i=1}^{n}\left[-0.5 * \log(2\pi\sigma^2) - \frac{\left(|\vec{B}|_i * F - |\vec{A}|_i * \cos(\theta_i)\right)^2}{2\sigma^2}\right] \qquad (7)$$

where $n$ is the number of residues (FL-WT, $n = 28$; FL-E76D, $n = 13$; FL-E76K, $n = 22$) and σ describes the general uncertainty of the model. To determine the best-fit values for $F$ and $\sigma$, Eq. (7) was maximized by setting its first derivative to zero and solving for the unknown parameters. Variances for $F$ and $\sigma$ were then approximated from the negative inverse Hessian matrix.

### Data availability
Structure factors and refined model of FL-C459E, ΔN-SH2, FL-E76D + SHP099, and FL-E76K + SHP099 have been deposited in the PDB under accession codes 6cmp, 6cmq, 6cmr, and 6cms, respectively. The NMR assignments of FL-WT, PTP, and tandem-SH2-E76K have been deposited in the BMRB under accession codes

27520, 27521, and 27522, respectively. The best model for FL-E76K based on the SAXS data was deposited in Model Archive under accession code ma-aks8e and the raw SAXS data in SASBDB under accession codes (FL-WT: SASDEN4; FL-E76K: SASDEP4). Other data are available from the corresponding author upon reasonable request.

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

## Acknowledgements

We would like to thank Susan S. Pochapsky for her assistance with the NMR experiments. This work was supported by the Howard Hughes Medical Institute and NIH (GM100966) to D.K. R.A.P.P was supported by the CNPq-Brazil Science Without Borders Fellowship [PDE 233809/2014-7]. The Berkeley Center for Structural Biology is supported in part by the National Institutes of Health, National Institute of General Medical Sciences, and the Howard Hughes Medical Institute. The Advanced Light Source is supported by the Director, Office of Science, Office of Basic Energy Sciences, of the U.S. Department of Energy under contract no. DE-AC02-05CH11231. Use of the Stanford Synchrotron Radiation Lightsource, SLAC National Accelerator Laboratory, is supported by the U.S. Department of Energy, Office of Science, Office of Basic Energy Sciences under contract no. DE-AC02-76SF00515. The SSRL Structural Molecular Biology Program is supported by the DOE Office of Biological and Environmental Research, and by the National Institutes of Health, National Institute of General Medical Sciences (including P41GM103393). The contents of this publication are solely the responsibility of the authors and do not necessarily represent the official views of NIGMS or NIH.

## Author contributions

D.K., Y.S., R.A.P.P., and I.M. conceived the project and designed the experiments; Y.S. performed and analyzed the NMR experiments with help from J.B.S. and R.O.; R.A.P.P. collected and analyzed the X-ray and SAXS data; I.M. and R.A.P.P. recorded and analyzed steady-state enzyme kinetics data and W.P. performed the steady-state simulations; W.P. and Y.S. performed and analyzed the stopped-flow experiments with help in sample preparation from I.M. and R.A.P.P.; Y.S., R.A.P.P., and I.M. performed the ITC experiments; all authors prepared figures and wrote the manuscript.

## Additional information

**Competing interests:** The authors declare no competing interests.

