## [Peer Review File · Nature Communications]

Reviewers' comments:

Reviewer #1 (Remarks to the Author):

This manuscript by Kern et al addresses a highly relevant and urgent topic of current biomedical cancer research namely the structures of mutated SHP2 and the potential of allosteric inhibitors of SHP2.

SHP2 has been discovered recently as an effective target for blocking the proliferation of cancers cells and inhibitors of the PTP domain as well as allosteric inhibitors have been successfully demonstrated in animal models.

In 2016 a Novartis group has published an allosteric inhibitor of the full-length protein. This molecule could be a valuable complementation of the orthosteric inhibitors earlier published earlier as an allosteric inhibitor might have advantages with respect to specificity which often has been an issue with orthosteric inhibitors.

Many SHP2-dependent cancers, however, contain mutations in the SHP2 gene and these mutations are usually located in the SH2-domains and therefore can hamper the mechanism of autoinhibition in the full-length protein. One potential disadvantage of allosteric inhibitors of SHP2 is they might not inhibit the clinically relevant mutated forms of SHP2.

The submitted manuscript investigates this issue for the most important (and most active) mutated SHP2, the E76K mutation, and for the allosteric SHP2-inhibitor SHP099.

The structure of the mutated SHP2 E76K was solved for a truncated version of the enzyme (without N-SH2) using protein crystallography (Fig 2). For the full-length mutated protein NMR spectra were recorded (Fig 1), however, without solving the structure. A comparison of the chemical shifts of the full-length mutated protein with those of the NMR of the truncated protein was then used to conclude that the chemical shifts observed in the NMR of the truncated structure were identical to those of the full-length structure, suggesting that the overall structure is not changed significantly when adding the N-SH2-domain.

Chemical shift differences within the PTP domain in the isolated PTP domain, the full-length mutants and the full-length wild type SHP2 were postulated to correspond to the respective equilibria between active (open) and inactive (closed) conformations and correlated with the enzymatic activities of the different proteins.

A crystal structure of the complex of mutated FL-SHP2 with SHP099 revealed that the inhibitor SHP099 binds to the closed (inactive) conformation of the protein thereby shifting the equilibrium to the closed conformation.

While this article certainly is a solid and also impressive piece of work that analyses the structures and the conformational flexibility of SHP2 and its thermodynamics / kinetics in great detail, the manuscript unfortunately remains hypothetical when it comes to the inhibitory effects of SHP099 on activated mutated or non-mutated proteins, which would be the biologically relevant target. The authors have recorded IC50 values only with the non-activated proteins (see Fig 6g-i), however, as seen in Fig 6h without using a native activating phosphopeptide ligand WT and mutant E76D are practically dead at these conditions. It remains unclear what an IC50 value of an essentially inactive protein is supposed to mean.

Another shortcoming of the article is that the contribution of N-SH2 to the active conformation remains unsolved as a truncated apo-structure of the mutant was crystallized.

Reviewer #2 (Remarks to the Author):

The manuscript by Kern and colleagues describes the structural implications of oncogenic mutations in the protein phosphatase SHP2 and how the overall conformation of these mutants is

modulated by the allosteric inhibitor SHP099. SHP2 is a potential drug target, and the inhibitor SHP099 was recently reported by Novartis. The study uses NMR, protein X-ray crystallography and biophysical methods such as enzyme kinetics and ITC to make the case about conformational selection to the closed, enzymatically inactive, conformation. Although I would have liked to see a complete study which also includes cellular studies, the experimental design, and data analysis of this paper is convincing. Overall, this is a very interesting paper that uses an impressive structural biology armament to pinpoint the molecular mode of action on SHP099 in clinically relevant mutant SHP2. However, some additional data and clarifications are needed to make a stronger case.

- The on- and off-rates of SHP099 in WT and mutant SHP2 should be given and discussed
- The chemical structure of the allosteric inhibitor should be shown in the manuscript
- The impact of allosteric vs. orthosteric phosphatase inhibitors should be discussed to give the reader a better insight
- The readers might not be familiar with the concept of conformational selection. Please explain and cited correctly O. F. Lange et al., Science 320, 1471 (2008)
- The term "healthy activities" (page 10) is misleading
- In the last paragraph of the results section, the authors try to give implications of kinetic differences for drug development. However, they do not finish their thoughts.
- It is well taken that E76D mutation shifts conformational equilibria. However, the authors should be more precise in depicting the actual atomic/ structural effect of this mutation. Maybe the authors find the term "charge inversion" useful for making the point.

Reviewer #3 (Remarks to the Author):

The manuscript "Mechanism of activating mutations and allosteric drug inhibition of the phosphatase SHP2" by Sun et al describes conformational and energetic investigations of the protein phosphatase, SHP2, with respect to mutations (e.g. E76D, E76K) and binding of the allosteric inhibitor, SHP099. The manuscript is well written and conveys clear messages, and the topic is of interest for a broad audience. I do, however, have some concerns about the technical quality of the manuscript.

1. The main results of this paper, such as the fraction of open versus closed SHP2 in wt and mutant protein, are based on NMR chemical shift arguments. This is not sufficient for a manuscript that claims "to dissect the energy landscape" of SHP2. NMR-based relaxation dispersion experiments should be carried out in order to see whether the conformational changes discussed in this paper fall within the rate regime of these experiments, in order to confirm the population of states, and in order to gain additional information such as exchange rates and WPD loop dynamics.
2. Small angle X-ray scattering (SAXS) experiments would complement and support the relevance of the C-SH2/PTP structure for full-length SHP2, and should be carried out with full-length SHP2 (and possibly C-SH2/PTP).
3. The results of enzyme kinetics should be presented quantitatively in a table, and compared to NMR-based conformational equilibria.
4. NMR resonance assignments should be deposited with the Biological Magnetic Resonance Bank.
5. There are a few typos that should be corrected (p.6/7 break should be Fig. 3a,b; p.10 "The K76D and K76E mutations" makes no sense)

If (and only if) the authors bring the technical quality of this manuscript to a state-of-the-art level by providing relaxation dispersion information and SAXS data, and improve the quality of the manuscript as suggested in items 3-5, I will support publication of this manuscript in Nature Communications.

Reviewer #1 (Remarks to the Author):

This manuscript by Kern et al addresses a highly relevant and urgent topic of current biomedical cancer research namely the structures of mutated SHP2 and the potential of allosteric inhibitors of SHP2. SHP2 has been discovered recently as an effective target for blocking the proliferation of cancers cells and inhibitors of the PTP domain as well as allosteric inhibitors have been successfully demonstrated in animal models.

In 2016 a Novartis group has published an allosteric inhibitor of the full-length protein. This molecule could be a valuable complementation of the orthosteric inhibitors earlier published earlier as an allosteric inhibitor might have advantages with respect to specificity which often has been an issue with orthosteric inhibitors.

Many SHP2-dependent cancers, however, contain mutations in the SHP2 gene and these mutations are usually located in the SH2-domains and therefore can hamper the mechanism of autoinhibition in the full-length protein. One potential disadvantage of allosteric inhibitors of SHP2 is they might not inhibit the clinically relevant mutated forms of SHP2.

The submitted manuscript investigates this issue for the most important (and most active) mutated SHP2, the E76K mutation, and for the allosteric SHP2-inhibitor SHP099.

The structure of the mutated SHP2 E76K was solved for a truncated version of the enzyme (without N-SH2) using protein crystallography (Fig 2). For the full-length mutated protein NMR spectra were recorded (Fig 1), however, without solving the structure. A comparison of the chemical shifts of the full-length mutated protein with those of the NMR of the truncated protein was then used to conclude that the chemical shifts observed in the NMR of the truncated structure were identical to those of the full-length structure, suggesting that the overall structure is not changed significantly when adding the N-SH2-domain.

Chemical shift differences within the PTP domain in the isolated PTP domain, the full-length mutants and the full-length wild type SHP2 were postulated to correspond to the respective equilibria between active (open) and inactive (closed) conformations and correlated with the enzymatic activities of the different proteins.

A crystal structure of the complex of mutated FL-SHP2 with SHP099 revealed that the inhibitor SHP099 binds to the closed (inactive) conformation of the protein thereby shifting the equilibrium to the closed conformation.

While this article certainly is a solid and also impressive piece of work that analyses the structures and the conformational flexibility of SHP2 and its thermodynamics / kinetics in great detail, the manuscript unfortunately remains hypothetical when it comes to the inhibitory effects of SHP099 on activated mutated or non-mutated proteins, which would be the biologically relevant target. The authors have recorded IC₅₀ values only with the non-activated proteins (see Fig 6g-i), however, as seen in Fig 6h without using a native activating phosphopeptide ligand WT and mutant E76D are practically dead at these conditions. It remains unclear what an IC₅₀ value of an essentially inactive protein is supposed to mean.

Another shortcoming of the article is that the contribution of N-SH2 to the active conformation remains unsolved as a truncated apo-structure of the mutant was crystallized.

We thank the referee for a thorough review and the endorsement of the manuscript. We have performed additional experiments and included new figures to the revised manuscript to fully address the two concerns raised:

1. We have now performed inhibition experiments in the presence of the activating phosphopeptide for WT, E76D, and E76K SHP2 and determined the requested IC₅₀ values for the activated proteins and

added these data to the manuscript (Supplementary Fig. 10). The results are in full agreement with our model that the inhibitor only binds to the closed conformation, thereby shifting this equilibrium to the closed form. In these experiments, the drug and the activating peptide are shifting the open/closed equilibrium in opposite directions and, therefore, the observed IC_{50} is the highest for E76K, followed by E76D and then WT. It is generally presumed that the increase in basal activity by these mutations is the primary source for cancer, and not the activity in the phosphopeptide-activated (or phospho-protein-activated *in vivo*) state for these mutant proteins. The activity of WT, E76D, and E76K mutant forms is identical in the presence of the activating phosphopeptide, further supporting the model that the mutations increase the basal activity by increasing the population of the open conformation. Consequently, the sole difference between WT and mutant forms is their respective activities in the absence of the phosphopeptide.

We have modified the text accordingly, including the discussion of the new inhibition data in the presence of the activating peptide shown in Supplementary Fig. 10.

2. We have performed SAXS experiments to further strengthen our conclusion in respect of the structural characterization of the active, open conformation of SHP2 (Supplementary Fig. 4). These results are very interesting and powerful since they fully buttress our main conclusions that were obtained from our NMR data on the full-length proteins combined with our X-ray structure of the ΔN -SH2 mutant: in the active conformation, the N-SH2 domain is detached from the rest of the protein, in sharp contrast to the inactive conformation. This is in full agreement with our NMR chemical shift data presented in Supplementary Fig. 3, which already showed nicely that the N-SH2 in the active conformation does not interact with the PTP domain. This finding differs from work published on the open conformation of the homologous protein SHP1 in which the C-SH2 and PTP domain are well superimposable to our ΔN -SH2 structure, but the N-SH2 domain interacts with the PTP domain. This raises the interesting question of whether this interaction captured in this X-ray structure is mainly due to crystal packing or whether the open forms of these two proteins in solution are indeed different. This could only be answered by performing similar NMR experiments applied here on SHP1, highlighting the power of solution NMR to shed light on such structural questions.

Reviewer #2 (Remarks to the Author):

The manuscript by Kern and colleagues describes the structural implications of oncogenic mutations in the protein phosphatase SHP2 and how the overall conformation of these mutants is modulated by the allosteric inhibitor SHP099. SHP2 is a potential drug target, and the inhibitor SHP099 was recently reported by Novartis. The study uses NMR, protein X-ray crystallography and biophysical methods such as enzyme kinetics and ITC to make the case about conformational selection to the closed, enzymatically inactive, conformation. Although I would have liked to see a complete study which also includes cellular studies, the experimental design, and data analysis of this paper is convincing. Overall, this is a very interesting paper that uses an impressive structural biology armament to pinpoint the molecular mode of action on SHP099 in clinically relevant mutant SHP2. However, some additional data and clarifications are needed to make a stronger case.

We thank the referee for the enthusiasm and support of this manuscript, and particularly for the insightful suggestions to make the paper stronger. We have edited the manuscript in response to all suggestions.

- The on- and off-rates of SHP099 in WT and mutant SHP2 should be given and discussed: These rates are given in Fig. 6 and discussed in the text.

- The chemical structure of the allosteric inhibitor should be shown in the manuscript: Yes, we added its chemical structure to Fig. 4 in panel a.
- The impact of allosteric vs. orthosteric phosphatase inhibitors should be discussed to give the reader a better insight:
We have expanded our discussion on this topic in the introduction and conclusion part.
- The readers might not be familiar with the concept of conformational selection. Please explain and cited correctly O. F. Lange et al., Science 320, 1471 (2008)
We added an explanation of the difference between conformational selection and induced fit in the text including the appropriate citations. The O.F. Lange is not a citation we elect because the flux was not measured in that study. The interpretation of conformational selection was solely routed in the existence of conformations that are very similar to the conformations seen in the bound complexes. While this criterion is a necessary condition for conformational selection, it is not a sufficient criterion to distinguish between the two opposing binding mechanisms (the flux of binding needs to be measured). Thank you for this suggestion to clarify this in the manuscript.
- The term "healthy activities" (page 10) is misleading: language changed
- In the last paragraph of the results section, the authors try to give implications of kinetic differences for drug development. However, they do not finish their thoughts: Thank you for this suggestion, we edited this paragraph to strengthen the conclusions!
- It is well taken that E76D mutation shifts conformational equilibria. However, the authors should be more precise in depicting the actual atomic/ structural effect of this mutation. Maybe the authors find the term "charge inversion" useful for making the point. Yes, we describe this now more precisely in the text together with Fig. 5f.

Reviewer #3 (Remarks to the Author):

The manuscript "Mechanism of activating mutations and allosteric drug inhibition of the phosphatase SHP2" by Sun et al describes conformational and energetic investigations of the protein phosphatase, SHP2, with respect to mutations (e.g. E76D, E76K) and binding of the allosteric inhibitor, SHP099. The manuscript is well written and conveys clear messages, and the topic is of interest for a broad audience. I do, however, have some concerns about the technical quality of the manuscript.

We thank the referee for the enthusiasm and support of this manuscript, and the raised questions/suggestions. We are happy to report that we have performed all suggested additional experiments, and they further confirm and strengthen the original conclusions. The revised manuscript also includes all other suggestions.

1. The main results of this paper, such as the fraction of open versus closed SHP2 in WT and mutant protein, are based on NMR chemical shift arguments. This is not sufficient for a manuscript that claims "to dissect the energy landscape" of SHP2. NMR-based relaxation dispersion experiments should be carried out in order to see whether the conformational changes discuss in this paper fall within the rate regime of these experiments, in order to confirm the population of states, and in order to gain additional information such as exchange rates and WPD loop dynamics.

We have performed NMR relaxation experiments for FL-SHP2 as suggested. No exchange contribution can be detected (see figure below) except in a few loops (data not shown). This is in agreement with the microscopic rate constants for opening and closing extracted from SHP099 stopped-flow fluorescence experiments (Fig. 6) that predicts the exchange to be in the slow time regime (see simulations below). Flat relaxation dispersion profiles are also seen for the WPD loop because as we had shown from our crystallographic analysis, binding of the N-SH2 into the active site prohibits movement of this loop. We can however confirm the slow opening/closing for WT SHP2 and the populations by a long [¹H-¹⁵N]-TROSY-HSQC spectrum that was performed on perdeuterated FL-WT. Additional peaks were found for

both N-SH2 and C-SH2 resonances along domain interfaces. We added a new figure (Supplementary Fig. 9) and a description of the new NMR data in the text.

CPMG experiments are insensitive to slow dynamics in SHP2 FL-WT and FL-E76K. (a) Kinetic schemes for FL-WT with microscopic rate constants for opening and closing extracted from SHP099 stopped-flow fluorescence experiments. (b) Theoretical exchange contribution (R_{ex}) for each residue was calculated with the Carver-Richards equation for two site exchange using microscopic rate constants from (a) and chemical shift differences between purely open (PTP/tandem-SH2) and purely closed (FL-WT + SHP099). As can be seen, the exchange is in the slow time regime and, therefore, only a small exchange contribution is produced ($R_{\text{ex}} \sim$ rate from major to minor). (c) Representative profiles of FL-WT ^{15}N CPMG relaxation dispersion experiments. While these residues have significant differences in chemical

shift between open and closed forms (Fig. 1b), no dispersion is observed as the exchange contribution is less than the uncertainty in $R_{2,\text{eff}}$ calculated from the peak intensities. (d) Zoom in of the X-ray crystal structure of auto-inhibited SHP2 (5EHP) with WPD loop residues shown in spheres. (e) ^{15}N CPMG relaxation dispersion profiles for WPD loop residues (excluded residues were overlapping). While the individual PTP domain possesses significant dynamics within the WPD loop as nicely demonstrated previously by Whittier *et al.*, binding of the N-SH2 into the active site prohibits this loop movement and results in flat dispersion profiles.

2. Small angle X-ray scattering (SAXS) experiments would complement and support the relevance of the C-SH2/PTP structure for full-length SHP2, and should be carried out with full-length SHP2 (and possibly C-SH2/PTP).

We have now performed the suggested SAXS experiments of full-length WT and full-length E76K to complement our findings from solution NMR and X-ray crystallography (Supplementary Fig. 4). Importantly, these new results fully confirm our findings by solution NMR, that E76K primarily samples an open conformation in which the N-SH2 is detached from the PTP domain. We added a paragraph in the text together with a figure (Supplementary Fig. 4).

3. The results of enzyme kinetics should be presented quantitatively in a table, and compared to NMR-based conformational equilibria.

We have added this table to Supplementary Fig. 8b. The NMR-based conformational equilibria are given in the text.

4. NMR resonance assignments should be deposited with the Biological Magnetic Resonance Bank. Yes, fully agreed, they are all deposited.

5. There are a few typos that should be corrected (p.6/7 break should be Fig. 3a,b; p.10 "The K76D and K76E mutations" makes no sense) Thank you, corrected.

If (and only if) the authors bring the technical quality of this manuscript to a state-of-the-art level by providing relaxation dispersion information and SAXS data, and improve the quality of the manuscript as suggested in items 3-5, I will support publication of this manuscript in Nature Communications.

We thank the reviewer for the insightful suggestions and we are happy that the additional experiments suggested further strengthened our main conclusions.

REVIEWERS' COMMENTS:

Reviewer #3 (Remarks to the Author):

By performing the suggested experiments, the authors have addressed all my concerns and suggestions. This is now an excellent paper, and I fully support its publication.

Reviewer #4 (Remarks to the Author):

The inclusion of scattering data does make a significant impact on the quality of the manuscript and also provides confidence in the open vs closed solution states of wt vs E76K.

The data itself looks very good and sup. Fig. 4. shows that significant differences exist between the WT and E76K states. Several small additions should be made to ensure the robustness of the analysis and support the structural conclusions:

1. Please provide a comparative figure of the real-space distance distributions. This should demonstrate to the reader that the open and closed conformations are observed in solution and are clearly different.
2. A table of SAXS parameters following the recommended standard is required (see: Trewthella, Jill, et al. "2017 publication guidelines for structural modelling of small-angle scattering data from biomolecules in solution: an update." Acta Crystallographica Section D 73.9 (2017): 710-728.).

It would be useful to mention observed differences in R_g and D_{max} in the text when discussing the SAXS data.

3. Please, if appropriate, find a common display in sup. fig. 4. for the models. It is hard for the reader to appreciate the differences in the models when panel d. has a unique orientation relative to panels b. and f.

4. SAXS data and models should be uploaded to the SASBDB (scattering data database constructed in collaboration with the PDB: www.sasbdb.org)

We are delighted that the referees are satisfied with our revisions and are of the opinion that our manuscript is now suitable for publication in Nature Communications. We have addressed their remaining points below and highlighted the changes in the manuscript using “track changes”.

Reviewer #3

By performing the suggested experiments, the authors have addressed all my concerns and suggestions. This is now an excellent paper, and I fully support its publication.

We thank reviewer 3 again for his constructive comments and are pleased to see that our revised manuscript addressed all the initial concerns.

Reviewer #4

The inclusion of scattering data does make a significant impact on the quality of the manuscript and also provides confidence in the open vs closed solution states of wt vs E76K.

The data itself looks very good and sup. Fig. 4. shows that significant differences exist between the WT and E76K states. Several small additions should be made to ensure the robustness of the analysis and support the structural conclusions:

We thank the reviewer for his positive evaluation of our manuscript and value the feedback regarding the SAXS data. All the suggestions have been incorporated as detailed below in our revised manuscript.

1. Please provide a comparative figure of the real-space distance distributions. This should demonstrate to the reader that the open and closed conformations are observed in solution and are clearly different.

A figure showing the real-space distance distributions for wild-type, full-length SHP2 and the E76K mutant form has been added as panel g in Supplementary Figure 4. It indeed clearly demonstrates that the open and closed conformations are different in solution; we thank the reviewer for this excellent suggestion.

2. A table of SAXS parameters following the recommended standard is required (see: Trewhella, Jill, et al. "2017 publication guidelines for structural modelling of small-angle scattering data from biomolecules in solution: an update." Acta Crystallographica Section D 73.9 (2017): 710-728.).

A table with SAXS parameters has been added as Supplementary Table 5. In the interest of brevity, we have opted not to duplicate information that is already present in the Methods section of the manuscript.

It would be useful to mention observed differences in R_g and D_{max} in the text when discussing the SAXS data.

We note that the paragraph comparing the SAXS data for FL-WT and FL-E76K (lines 224 – 236) does already discuss the differences in R_g and D_{max} .

3. Please, if appropriate, find a common display in sup. fig. 4. for the models. It is hard for the reader to appreciate the differences in the models when panel d. has a unique orientation relative to panels b. and f. While we agree with the reviewer that common orientation of the SAXS envelopes and structures would be useful, nevertheless it has proven impossible to find one that clearly shows all the features. Therefore, we have opted to keep panels b, d, and f in Supplementary Figure 4 unchanged.

4. SAXS data and models should be uploaded to the SASBDB (scattering data database constructed in collaboration with the PDB: www.sasbdb.org).

The SAXS models were deposited already in ModelArchive; as requested we have now deposited the raw data as well in SASBDB. The accession codes have not been assigned yet, but the entries will be released upon publication and the accession codes will be added at a later stage in the revision process.